evolution, plant science

bee pollination, *Clarkia*, Darwin, heteranthery, pollen presentation, intrasexual selection

**Author for correspondence:**
Kathleen M. Kay
e-mail: kmkay@ucsc.edu

# Darwin's vexing contrivance: a new hypothesis for why some flowers have two kinds of anther

Kathleen M. Kay, Tania Jogesh, Diana Tataru and Sami Akiba

Department of Ecology and Evolutionary Biology, University of California, 130 McAllister Way, Santa Cruz, CA 95060, USA

KMK, 0000-0001-8858-110X; TJ, 0000-0001-9092-0019; DT, 0000-0003-2632-2262; SA, 0000-0003-0807-7750

Heteranthery, the presence of two or more anther types in the same flower, is taxonomically widespread among bee-pollinated angiosperms, yet has puzzled botanists since Darwin. We test two competing hypotheses for its evolution: the long-standing 'division of labour' hypothesis, which posits that some anthers are specialized as food rewards for bees whereas others are specialized for surreptitious pollination, and our new hypothesis that heteranthery is a way to gradually release pollen that maximizes pollen delivery. We examine the evolution of heteranthery and associated traits across the genus *Clarkia* (Onagraceae) and study plant–pollinator interactions in two heterantherous *Clarkia* species. Across species, heteranthery is associated with bee pollination, delayed dehiscence and colour crypsis of one anther whorl, and movement of that anther whorl upon dehiscence. Our mechanistic studies in heterantherous species show that bees notice, forage on and export pollen from each anther whorl when it is dehiscing, and that heteranthery promotes pollen export. We find no support for division of labour, but multifarious evidence that heteranthery is a mechanism for gradual pollen presentation that probably evolved through indirect male–male competition for siring success.

## 1. Introduction

Bee pollination poses an intriguing conflict for plants; although plants rely on bees to transfer their pollen for sexual reproduction, the bees collect pollen to provision their nests. Plants should thus experience selection for mechanisms that promote pollen export to stigmas while reducing pollen consumption by bees. One proposed mechanism is heteranthery, in which anthers of different colour, size and position occur within the same flower.

Heteranthery is taxonomically widespread among bee-pollinated plants and has a long history of investigation [1]. Darwin spent many decades documenting flower-pollinator fit and identifying floral traits that would promote effective outcrossing [2,3] (reviewed in [4]). Yet the phenomenon of heteranthery perplexed him; he famously wrote to J. D. Hooker regarding heterantherous plants, 'I am very low about them, and have wasted enormous labour over them, and cannot yet get a glimpse of the meaning of the parts' [5]. Müller [6,7], having recognized the conflict of interest intrinsic to pollen removal by bees, hypothesized that heteranthery functions as a division of labour strategy in which some anthers primarily make pollen for plant reproduction whereas others make pollen for bee attraction and consumption. Under this hypothesis, bees are attracted to the feeding anthers during a flower visit, while pollinating anthers surreptitiously place pollen on the bee body in a location likely to escape grooming and promote pollen export [6,8,9]. Flowers categorized as heterantherous typically present a suite of traits suggesting division of labour, including centrally located and visually conspicuous 'feeding anthers' and peripherally deflected inconspicuous 'pollinating anthers' [1,4,7,10]. Support for division of labour

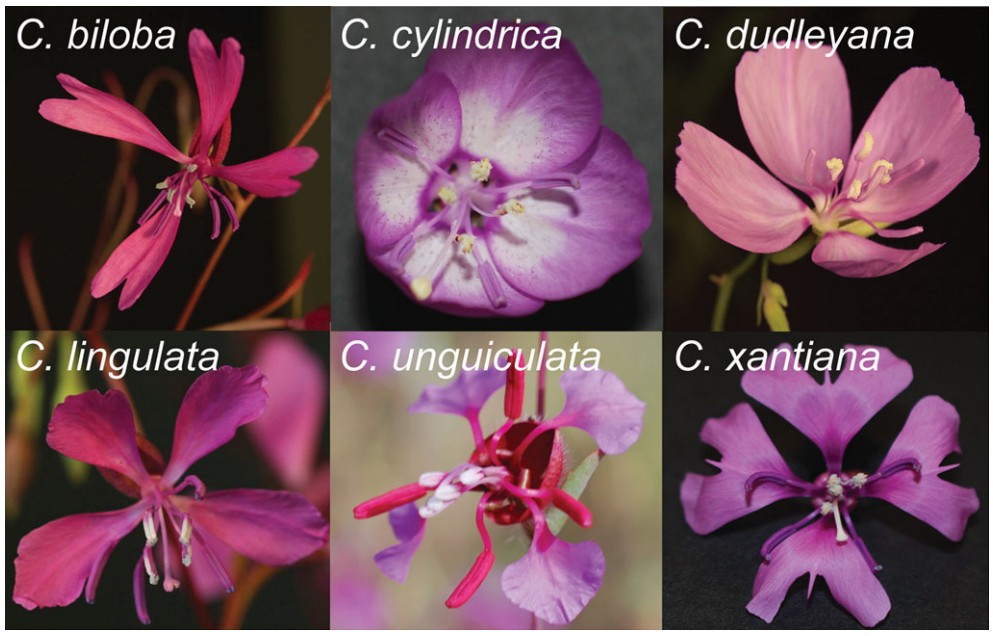

**Figure 1.** A selection of *Clarkia* flowers exhibiting heteranthery. Flowers are recently opened, showing the erect conspicuous inner anther whorl and the inconspicuous and reflexed pink, purple or red outer anther whorl. Photos by K. Kay and D. Tataru.

has been demonstrated in *Solanum rostratum*, in which several bright yellow central anthers are preferentially handled by bees while a single dull anther is deflected to the side and exports more pollen to stigmas [11]. However, in the few other heterantherous species that have been investigated, support for division of labour between feeding and pollinating anthers has been equivocal [12–16]. Nevertheless, division of labour is the only explanation of heteranthery put forth to date.

Here we propose and test an alternative hypothesis—that heteranthery is a gradual pollen presentation strategy for reducing pollen loss to bees while increasing pollen export and siring success. We test this hypothesis in *Clarkia* (Onagraceae), a primarily bee-pollinated genus. Many *Clarkia* exhibit heteranthery in the form of colour, size and positional differences between two anther whorls (figure 1). *Clarkia* flowers appear to have quintessential 'pollinating' versus 'feeding' anthers, with visually conspicuous central anthers and inconspicuous peripheral anthers. Yet we noticed that heterantherous species tend to release pollen from inner and outer anther whorls at different times and rates, making division of labour between feeding and pollinating functions unlikely, since the putative pollinating anthers are not yet mature when the feeding anthers would be attracting and rewarding bees. Moreover, in the only investigation of heteranthery in *Clarkia*, pollen from the inconspicuously coloured outer anthers, which putatively specialize on pollinating, was found to pollinate more poorly, in terms of stigma penetration and pollen tube growth [17]. Instead of dividing labour between pollinating and feeding functions during a bee visit, we propose that the inconspicuous colour and deflection of the outer anther whorl make it cryptic until the conspicuous inner whorl has dehisced, at which point the outer anthers move to the center of the flower and gradually release pollen. Gradual pollen release could reduce pollen lost to grooming and wastage and subsequently increase total pollen export and male fitness across many bee visits, as has been shown for other floral traits that stagger the release of pollen, such as gradual anther, flower, or inflorescence maturation [18,19] (reviewed in [20,21]). Heteranthery may thus represent a previously unidentified mechanism of pollen dosing.

We conduct a series of studies in *Clarkia* to characterize traits associated with heteranthery, to simultaneously test the longstanding division of labour hypothesis and our alternative explanation of gradual pollen presentation, and to explore the reproductive benefits of pollen dosing by heteranthery. We first ask whether heteranthery is consistently associated with bee pollination and delayed pollen release from one anther whorl across *Clarkia*. Both hypotheses predict an association with bee pollination; however, division of labour predicts that the two anther whorls release pollen simultaneously, whereas pollen dosing predicts gradual and delayed dehiscence of the outer anthers. Second, we examine the fate of pollen from both anther whorls in a natural population of heterantherous *C. cylindrica*. Division of labour predicts that bees primarily collect pollen from inner anthers and export pollen from outer anthers. In contrast, pollen dosing predicts that both types of pollen are collected by bees and both types are exported to stigmas. Third, we examine crypsis of the outer anther whorl with multiple approaches. Division of labour predicts that the inconspicuous outer anther whorl should be consistently cryptic to evade pollen collection by bees, whereas pollen dosing predicts the outer anthers should be targeted by bees once they have dehisced. We quantify the visibility of flower parts across several *Clarkia* species using reflectance data. We then test responses of native bee pollinators to the putative crypsis by quantifying bee visitation when each whorl of anthers is removed. In captive bumblebee trials, we further explore the outer anther crypsis by factorially manipulating colour and position. Finally, we test the reproductive benefits of outer anther pollen dosing by measuring whether bumblebees export more pollen from *Clarkia* with gradually dehiscing outer anthers than from those with simultaneous pollen release.

## 2. Material and methods

### (a) Study system and field sites
*Clarkia* is a genus of annual herbs mostly endemic to the California Floristic Province [22]. Although all species are self-compatible,

most are predominantly outcrossing with prominent herkogamy and protandry, whereas others are exclusively self-fertilizing [23,24]. The heterantherous species all have obvious differences in anther colour between the two whorls of four anthers each. Anthers dehisce lengthwise from tip to base (electronic supplementary material, movie S1).

We conducted field studies in Pinnacles National Park in the inner coast range of California. Pinnacles National Park is characterized by cool wet winters and intensely hot summers, and *Clarkia* blooms at the transition between seasons. The *C. cylindrica* population was located at Bear Gulch Canyon (36.4828 N, 121.1668 W; WGS 1984), and the population of *C. unguiculata* was found along Hwy 146 and the Bench Trail near the turn off to the Bear Gulch Day Use Area (36.4858 N, 121.1675 W). Both species are primarily pollinated by oligolectic solitary bees (primarily *Hesperapis regularis*; Melittidae), with additional visits from bumblebees and honey bees [25], all of which forage for pollen and nectar.

Greenhouse and flight cage studies were conducted in the University of California, Santa Cruz greenhouse facilities. We planted seeds in 3.8 cm diameter cone-tainers (Stuewe and Sons, Inc.) with a 4 : 1 potting soil (Pro-Mix HP Mycorrhizae) to perlite mix. These seeds were germinated in Conviron E-15 growth chambers with a 15°C, 10 h day/10°C, 14 h night schedule and watered with diH$_2$O every other day. Once seedlings had secondary leaves, we transferred them to the greenhouse, where they were kept between 13 and 25°C with a 13.5 h day and daily overhead watering. Plants for the comparative study were planted in Fall 2017 and plants for bumblebee trials were planted in spring of 2018 and winter of 2019.

## (b) Is heteranthery associated with bee pollination, anther movement and anther timing differences across *Clarkia*?

We used the consensus phylogeny for diploid *Clarkia* species in Briscoe *et al.* [26] to reconstruct the evolutionary history of heteranthery, in this case strictly defined as a colour difference between the inner and outer anther whorls, and to test whether it shows correlated evolution with bee pollination, outer anther movement and anther timing differences. To characterize heteranthery, we reviewed photos from CalPhotos (https://calphotos.berkeley.edu), descriptions from the Smithsonian Onagraceae taxonomy website (https://naturalhistory2.si.edu/botany/onagraceae), and descriptions in the Jepson manual [27]. We scored pollination system as a binary trait: either predominant bee pollination or not. Other forms of pollination include autogamy, fly and hawkmoth. We gathered pollination data from the literature (electronic supplementary material, table S1). Finally, we used CalPhotos and our own time lapse photos to score outer anther movement. Species either have consistently erect outer anthers or they have outer anthers that are reflexed in newly opened flowers and become erect as the flower ages. Anther movement was scored as a binary trait by comparing photos of flowers at different ages.

To assess quantitative differences in anther timing, we grew 11 *Clarkia* species in the greenhouse and took time lapse photos of flower maturation. We included six species with heteranthery and five species without (electronic supplementary material, table S1). Greenhouse-grown plants were brought into the lab and staged with 24 h standardized lighting and a ceramic heat lamp on a 12 h on-off schedule. We took photos of one flower per species every 10 min with a digital SLR camera over the course of anthesis, which ranged from 2–5 days. For *C. concinna* and *C. amoena*, we were unable to complete photo series, and these were assessed visually every 12 h for anther timing. Timing separation was calculated as one minus the number of photos (or visual checks) in which both anther whorls were actively exposing pollen divided by the number of photos (or visual checks) in which any anthers were

actively exposing pollen. This index ranged from zero for simultaneous dehiscence to one for complete separation. *Clarkia breweri* and *C. concinna* only have one whorl of simultaneously dehiscing anthers and were assigned a separation index of zero.

We reconstructed the history of heteranthery across *Clarkia* using stochastic character mapping. To determine whether heteranthery is associated with bee pollination or outer anther movement, we used Pagel's [28] test for correlated evolution of two binary traits on the randomly resolved phylogeny. For heteranthery and pollination system, we compared three models with AIC: one in which transition rates for both pollination and heteranthery vary with the state of the other character, one in which transitions in pollination vary with the heteranthery character state and one in which transitions in heteranthery vary with the pollination character state (our hypothesis). These models were tested against a null model in which the transition rates of both characters (heteranthery and pollination) vary independently of the other character state. We used an analogous series of models to test for correlated evolution between heteranthery and outer anther movement (electronic supplementary material, appendix S1). We then tested whether anther timing separation, as measured by our index score, differs between heterantherous and non-heterantherous species using Garland *et al.*'s [29] phylogenetic ANOVA. All comparative analyses were implemented with the phytools R package [30].

## (c) Does one set of anthers feed bees and the other export pollen, as predicted by division of labour?

We followed the fate of pollen from inner versus outer anther whorls of *C. cylindrica* in the field. Flowers have four pale pinkish-purple, cup-like petals and two anther whorls (figure 1). Inner anthers appear whitish-yellow and produce white pollen, whereas outer anthers are pinkish-purple and produce purple pollen. This consistent colour difference allowed us to directly address the division of labour prediction that the inner anthers predominantly reward bees, whereas the outer anthers predominantly export pollen to stigmas.

We sampled bees and stigmas in the field in the spring of 2017 during peak flowering. We set out a white bowl of soapy water to trap bees at each of four locations approximately 30 m apart. Bowls were exposed to bees in the morning from 08.00 to 12.00 and in the afternoon from 12.30 to 16.30 over the course of 2 days. At the end of each time period, we collected all the bees and 5–6 open stigmas from the area near each bee bowl. Bees and stigmas were frozen until pollen counting and pinning occurred. Bees were later determined to be *Hesperapis regularis*, a solitary melittid bee specialized on *Clarkia* [25]. Under a dissecting microscope, we counted both white and purple *Clarkia* pollen in the bees' scopae, which is destined to provision nests, and on the stigmas, which indicates actual pollen export. *H. regularis* glazes scopal loads with nectar so that they are densely clumped [31]. We removed the pollen from one scopa per bee and mixed it into glycerol on a gridded slide. In *C. cylindrica*, the outer anthers are longer than the inner anthers, so we counted pollen on greenhouse-grown plants to generate a null expectation for the ratio of pollen types. We counted pollen from one inner and one outer anther from seven individual plants. We then tested whether stigmas and bee scopae differ in their proportions of pollen types. We analysed the proportion of purple pollen using a linear model with sample source (scopa versus stigma) and total pollen count as explanatory variables.

## (d) Are outer anthers cryptic, and do they become conspicuous to bees as they release pollen?

Our pollen dosing hypothesis predicts that outer anthers are initially cryptic but become apparent to bees as they release

pollen. We took a multi-pronged approach to this question, first analyzing colour reflectance across many heterantherous species, then manipulating the presence of flower parts in the field for one species, and finally manipulating the colour and position of outer anthers with captive bumblebees.

### (i) Colour reflectance

We used an Ocean Optics JAZ spectrometer to measure UV-VIS reflectance of individual flower parts for a representative freshly opened flower from six heterantherous species of *Clarkia*. We used a full spectrum light source (Ocean Optics PX-2, pulsed xenon lamp 220–750 nm) and the following settings: 10 millisecond integration time, boxcar width of 5, 50 averaged scans per reading, and 45° probe angle. We scaled measurements between a white standard (Ocean Optics WS-1-SL) and a dark sample. To model how the colours would be seen by bees, we used characteristics of the *Apis mellifera* visual system, including receptor noise ratios [32] and photoreceptor sensitivity wavelengths [33] with the package pavo 2 [34] and custom scripts in R. Although *Clarkia* are rarely visited by honey bees, spectral sensitivities are similar across Hymenoptera [33,35]. To compare flower parts, we used modelled green contrast values (contrasts between a stimulus and its background mediated by the green photoreceptor), which have been shown to accurately predict honey bee visual detection and pattern processing, especially with small flowers [36]. We used phylogenetic paired *t*-tests implemented in phytools to compare pollen from the inner anthers to undehisced outer anthers, which are both visible on fresh flowers. We also compared inner pollen with stigmas because we suspected they were similarly apparent to bees.

### (ii) Anther removal in the field

To understand how bee vistation responds to each whorl of anthers, we conducted an anther removal field experiment on *C. unguiculata* during peak flowering in 2017. Whereas division of labour predicts that bees preferentially forage on the conspicuous inner anther whorl, our pollen dosing hypothesis predicts that bees forage on whichever anthers are actively dehiscing. We set up patches with five different treatments on freshly opened flowers, including inner anther removal, outer anther removal, pistil removal, inner anther plus pistil removal, and a control in which we similarly handled and labelled the flowers but did not remove anything (electronic supplementary material, figure S1). We included pistil removals because the pistil is similarly coloured to the inner anther whorl and may similarly affect bee visitation. Subtending stems were labelled with red tape, with two flowers per treatment per patch. We observed each of 17 patches for 20 min periods twice during the morning when only the inner anthers had exposed pollen, and twice during the afternoon when the outer anthers had moved toward the center of the flower and begun to dehisce. We repeated these treatments on fresh flowers for 2 days. We analyzed visitation counts separately during the morning and afternoon with generalized linear mixed models with a Poisson error distribution, a fixed treatment effect and a random patch effect using the lme4 R package [37].

Our study design also allowed us to test one possible alternative explanation for gradual pollen dehiscence—specifically, that pollen released at the end of the male phase could assure seed set through pollinator attraction and/or autogamy during female phase. We considered this explanation unlikely because stigmas open 2–7 days after outer anther dehiscence begins, at which time pollen is typically no longer present on the shriveled anthers, and styles elongate beyond the anthers at the time of receptivity [17]. Nevertheless, we returned three weeks later to harvest ripe fruits from our treatment flowers. We compared seed set and the percentage of ovules fertilized with one-way ANOVAs across the treatments of inner anthers removed, outer anthers removed, and control.

### (iii) Anther colour and position manipulations

We tested whether the inconspicuous colour and initial reflexed position of the outer anthers in *C. unguiculata* reduced bee access to pollen, beyond any reduction provided by delayed dehiscence. In 2019, we planted approximately 300 *C. unguiculata* from Pinnacles National Park under standard greenhouse conditions (see above). We purchased a *Bombus impatiens* colony (Arbico Organics) and provided worker bees access to *C. unguiculata* flowers for the first 3 days to accustom them to a new food source. The colony also had access to a sugar solution. Although *B. impatiens* is not a native *Clarkia* pollinator, it readily collects pollen and nectar from *Clarkia* flowers, and visual sensitivities are similar across all bees [33,35]. After the training period, we exposed bees to arrays of 15 freshly male *Clarkia* flowers in a 0.9 m × 0.6 m × 0.6 m pop-up insect cage with a viewing window on one side (Bioquip). We removed inner anthers from all flowers to isolate bee responses to outer anthers. Each array was randomly assigned to one of four treatments that factorially altered the colour and position of outer anthers. We painted the filament and anther base (away from the exposed pollen at the tip) with either inconspicuous red paint or conspicuous yellow paint (Blick matte acrylic paint, red deep and yellow bright, respectively; electronic supplementary material, figure S2). We altered anther position by either loosely tying cotton thread (painted the same colour as the filaments) around the base of the outer anther whorl to leave them reflexed or by cinching the thread to make them erect. We allowed a single bee to enter the cage directly from the hive. Once it began foraging, we unzipped the door to the cage so that the bee could freely leave. For each array, we recorded the bee action (nectaring or collecting pollen) during the first flower visit, the proportion of visits that involved pollen collecting and the total number of nectaring visits. We used $\chi^2$ tests to compare the first visit type across colour and position categories separately. We used two-way ANOVAs with colour and position as predictors to examine other response variables. We completed 51 arrays, spread approximately evenly across treatments.

## (e) Does gradual pollen presentation increase pollen export?

We tested our hypothesis that pollen dosing by the cryptic outer anthers increases pollen export by manipulating pollen availability in a greenhouse experiment with *C. unguiculata* and captive bumblebees. In 2018, we grew approximately 300 *C. unguiculata* from Pinnacles National Park. We used the same captive *Bombus* setup as described above, except the floral arrays each held 20 flowers in male phase interspersed with five emasculated and receptive female phase flowers. We removed inner anthers from all flowers to isolate effects of the pollen release schedule of outer anthers. All male phase flowers in an array were one of two treatments: a 'dosing' treatment in which outer anthers were just beginning to dehisce at the tips and a 'no-dosing' treatment in which outer anthers were fully dehisced (electronic supplementary material, figure S3). Although flowers in the no-dosing treatment were 1 day older (out of a 4–6-day male phase in the greenhouse), they were still fully turgid and the pollen appeared to have the same consistency. We measured the mm of exposed pollen along the length of the gradually opening anthers on a subset of 20 flowers for each treatment to generate an expectation for the relative amount of pollen available for collection and export between array types. Once a bee left the array, we collected the five stigmas to count pollen under the dissecting microscope and collected the bee to weigh the pollen in one corbicula. Arrays were visited by a single bee, and bees and visited flowers were not reused. We

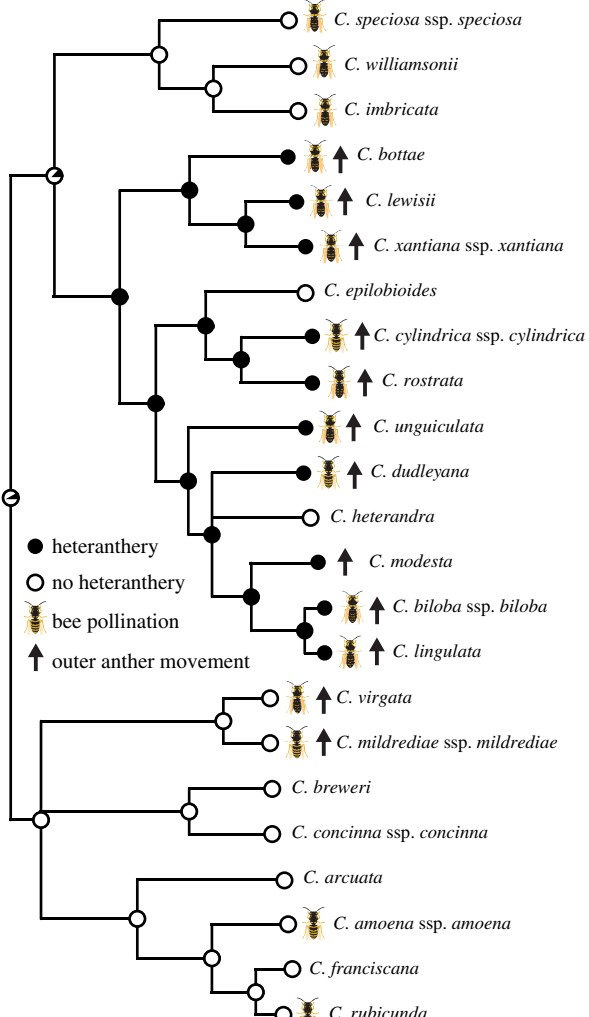

**Figure 2.** The consensus phylogeny of diploid *Clarkia* shows a single evolution of heteranthery in a predominantly bee-pollinated clade. There are two independent losses of heteranthery in autogamous species (*C. epilobioides* and *C. heterandra*, the latter of which has sterile inner anthers), whereas one autogamous species (*C. modesta*) retains both heteranthery and anther movement. (Online version in colour.)

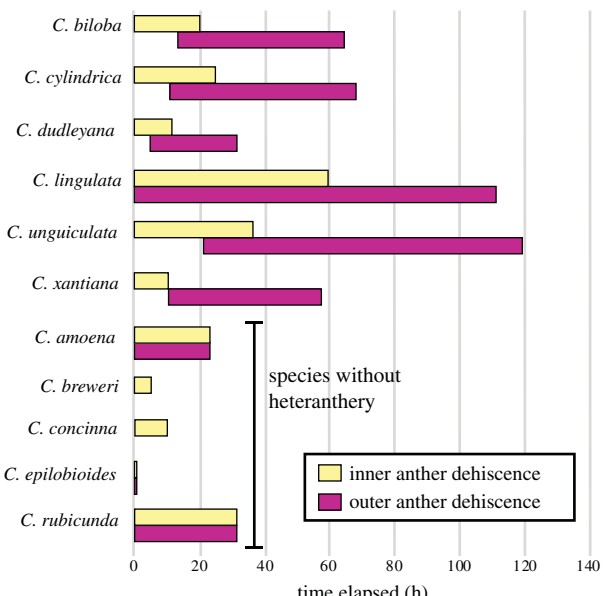

**Figure 3.** Timing and duration of inner and outer anther dehiscence since anthesis for six *Clarkia* species with heteranthery (top) and five without (bottom), as determined by time lapse photography. *Clarkia breweri* and *C. concinna* only have one anther whorl, and *C. epilobioides* anthers fully dehisce prior to flower opening. Heterantherous species have significantly less overlap between anther whorls in the timing of pollen release. (Online version in colour.)

compared stigma pollen counts between treatments with a negative binomial generalized linear mixed model with treatment as a fixed effect and array as a random effect in the lme4 package in R. We compared the weight of pollen collected in corbicula between treatments with a one-way ANOVA. We completed 13 dosing arrays and 15 no-dosing arrays.

## 3. Results

### (a) Is heteranthery associated with bee pollination, anther movement and anther timing differences across *Clarkia*?

Heteranthery probably evolved once in the genus *Clarkia*, with two subsequent losses, and is commonly present with both bee pollination and anther movement (figure 2). All heterantherous species except one are bee pollinated, whereas several bee-pollinated species do not have heteranthery. Accordingly, the best-fit Pagel model for trait evolution suggests that transitions in heteranthery are correlated with pollination state, although it is only a marginally better fit than character independence ($p = 0.055$; see electronic supplementary material,

appendix S1 for full details of all statistical models). All heterantherous species exhibit movement of the outer anthers from a reflexed to erect position as the flowers age, although two bee-pollinated species without heteranthery also show anther movement. The best-fitting Pagel model for correlated evolution of anther movement and heteranthery is that transitions in heteranthery are correlated with anther movement state ($p = 0.004$). Timing differences in anther dehiscence are ubiquitous for heterantherous species, and unobserved for any other species (phylogenetic ANOVA, $F = 75.71$, $p = 0.002$). Outer anther dehiscence typically starts after inner anther dehiscence and lasts substantially longer (figure 3; electronic supplementary material, movie S1).

### (b) Does one set of anthers feed bees and the other export pollen, as predicted by division of labour?

Pollen from both anther whorls of *C. cylindrica* is collected by bees in their scopae and exported to stigmas in similar proportions (figure 4; sample source: $t = -1.401$, $p = 0.17$). There is a positive relationship between total pollen in a sample and the proportion of purple pollen (slope = $1.943 \times 10^{-4}$, s.e. = $6.506 \times 10^{-5}$, $t = 2.986$, $p = 0.005$) across both sample types. Moreover, both stigmas and scopae averaged lower proportions of purple outer anther pollen than produced by flowers (mean = 0.61, SD = 0.03, $n$ = one anther of each whorl from seven plants); thus, the inner anther pollen is both collected and exported at a higher rate than the outer anther pollen, but both anther types perform both functions.

### (c) Are outer anthers cryptic, and do they become conspicuous to bees as they release pollen?
#### (i) Colour reflectance
Interpreted with a bee vision model, outer anthers are relatively colour cryptic, whereas inner anthers and stigmas are

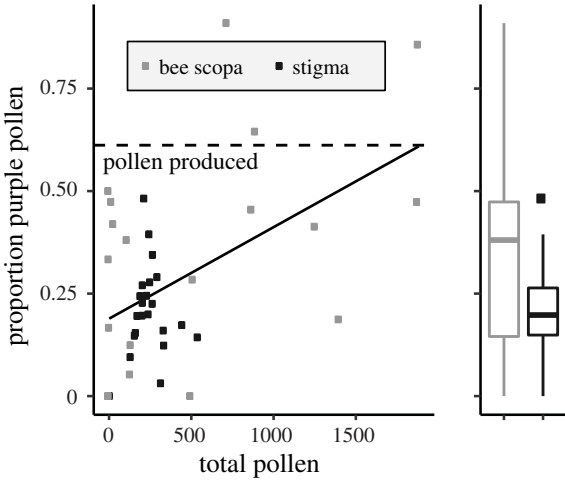

**Figure 4.** The proportion of purple (outer anther) pollen found on bee scopae and plant stigmas does not differ but does increase with total pollen counts in a natural population of *C. cylindrica*. Flowers produce a higher proportion of purple pollen (dashed line, estimated from greenhouse-grown plants) than is typically found on either stigmas or bees.

conspicuous, across six *Clarkia* species with heteranthery (figure 5). In particular, undehisced outer anthers have significantly lower modelled green contrast than inner pollen during the early stages of flowering ($t = 6.75$, d.f. = 3, $p = 0.007$). The inner pollen and stigma were similarly conspicuous ($t = 0.59$, d.f. = 3, $p$-value = 0.60), which led us to include pistil removal in our field experiment below.

### (ii) Field manipulations

The effects of removing anthers from freshly opened *C. unguiculata* flowers depend on the time of day. In the morning, when the outer anthers are reflexed and inconspicuous (figure 6a), removing the outer anthers has no significant effect, whereas removing the inner anthers or the inner anthers plus pistil decreases native bee visitation by 50% and 70%, respectively, compared to control flowers (figure 6b). In contrast, in the afternoon when the outer anthers have become erect and begun to dehisce, removing either whorl of anthers reduces visitation, by 30% for inner anthers and 45% for outer anthers. Removing just the pistil, which is highly conspicuous and similar in colour to the inner anthers, also decreases visitation in the afternoon when there is little inner pollen remaining.

Removing either anther whorl has no effect on total seed set or the percentage of fertilized ovules, countering the hypothesis that anthers function for attracting pollinators during female phase or for delayed selfing. Control flowers, flowers with inner anthers removed and flowers with outer anthers removed, respectively, averaged 34, 28 and 30 seeds ($F_{2,78} = 0.43$, $p = 0.65$), and 53%, 43% and 49% of ovules fertilized ($F_{2,78} = 0.89$, $p = 0.41$).

### (iii) Anther colour and position manipulations

In greenhouse trials with bumblebees and *C. unguiculata*, we show that the colour of the outer anthers *per se* can reduce initial bee visitation to the outer anthers and shift behavior towards nectaring instead of pollen collecting. The first behavior to a flower was more likely to be nectaring when the outer anthers were painted inconspicuously red, but pollen collecting when painted conspicuously yellow ($\chi^2 = 8.1$, $p = 0.004$).

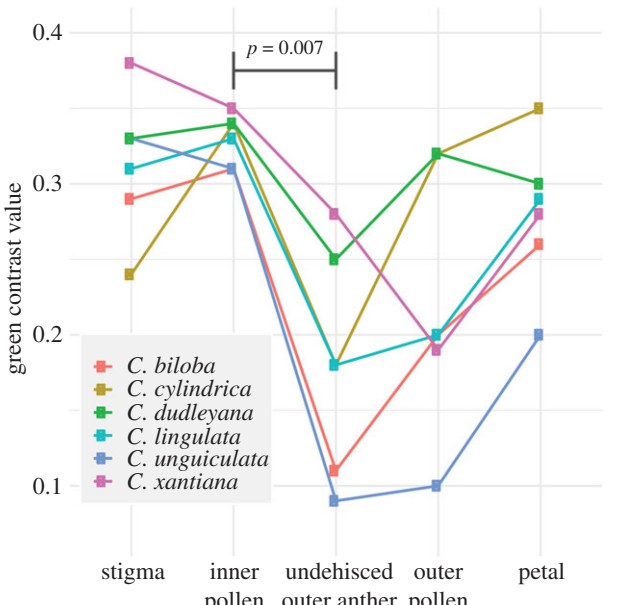

**Figure 5.** Green contrast values from flower colour reflectance from six heterantherous *Clarkia* species. In freshly opened flowers, undehisced outer anthers are significantly less conspicuous than the exposed pollen from inner anthers.

Across foraging bouts, red anther colour decreased the proportion of visits in which pollen collecting behavior was observed (LS means: 0.60 for red and 0.75 for yellow; entire model $F_{3,47} = 3.10$, $p = 0.04$; colour $p = 0.01$) and increased the total amount of nectaring visits during the foraging bouts (LS means: 20.3 for red and 9.4 for yellow; entire model $F_{3,47} = 2.6$, $p = 0.06$; colour $p = 0.027$). In contrast, anther position did not significantly affect any of our response variables, either alone or interacting with colour.

### (d) Does gradual pollen presentation increase pollen export?

In greenhouse trials with *C. unguiculata*, gradually dosing flowers exposed far less pollen from outer anthers but exported similar amounts to female phase flowers (figure 7). Completely dehisced flowers exposed approximately 2.5 times the amount of pollen as freshly opened unmanipulated flowers (4.6 mm versus 1.9 mm of exposed pollen, $F_{1,38} = 483.17$, $p < 0.001$). Nevertheless, stigmas in the completely dehisced arrays received about 30% less pollen than in the gradually dosing arrays. Although this difference was not statistically significant (log odds ratio for no-dosing treatment: −0.3507, $p = 0.27$), it is in the opposite direction than expected based on pollen availability. At the end of their foraging bouts, bumblebees had collected similar amounts of pollen in their corbiculae when visiting each treatment (dosing treatment: 6.7 mg, no-dosing treatment: 6.5 mg; $F_{1,25} = 0.01$, $p = 0.923$).

## 4. Discussion

In *Clarkia*, multifarious evidence contradicts division of labour between pollinating and feeding functions of anthers but supports our alternative hypothesis that heteranthery is a pollen dosing mechanism. Heteranthery probably evolved once in a predominantly bee-pollinated clade and is consistently associated with slow and delayed pollen release from the cryptically

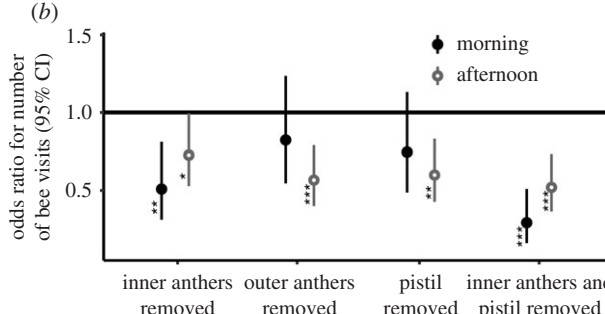

*(a)*

*C. unguiculata* (morning)    *C. unguiculata* (afternoon)

*(b)*

**Figure 6.** (*a*) Field photos (K. Kay) of *C. unguiculata* flowers used as experimental controls. In the morning, newly opened flowers had inner anthers beginning to release pollen and outer anthers reflexed, whereas in the afternoon inner anthers were shriveled and outer anthers were erect and releasing pollen. (*b*) Proportional reduction in bee visitation compared to control flowers (horizontal black line) for altered *C. unguiculata* flowers in the morning (closed circles) and afternoon (open circles; *p < 0.05, **p < 0.01, ***p < 0.001). Photos of modified flowers are found in electronic supplementary material, figure S1. (Online version in colour.)

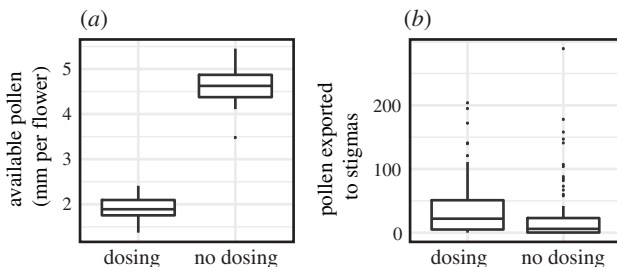

**Figure 7.** Although dosing *C. unguiculata* flowers expose far less pollen than fully dehisced flowers (*a*), similar amounts are exported to stigmas (*b*).

coloured outer anther whorl and movement of the outer anthers toward the center of the flower upon dehiscence. Staggered dehiscence between anther types suggests heteranthery is a way for the flowers to gradually present their pollen to bees.

In a natural population of *C. cylindrica*, pollen from both anther whorls is collected by bees and transferred to stigmas in similar proportions. These results directly contradict the division of labour prediction that different anthers specialize on pollen export versus pollinator reward functions. Outer anther pollen is underrepresented on both bees and stigmas, relative to its higher rate of production. We hypothesize that outer anther pollen is subject to a higher risk of loss by florivores and wind simply because it is present on the flower longer. Exposure to the elements, such as ultraviolet light, may also explain the slight decrease in performance found in outer anther pollen in *C. unguiculata* [17]. Whereas the greater disappearance and lower siring success of outer anther pollen are puzzling, neither supports outer anthers being specialized on pollination versus feeding.

We additionally study whether the outer anthers are cryptic and whether the crypsis lessens as the outer anthers dehisce

and become erect. Our colour reflectance data from multiple heterantherous species suggest that outer anthers are less apparent to bees than inner anthers. In the field, bees initially reduce visitation to freshly opened *C. unguiculata* flowers lacking conspicuous inner anthers but not those lacking outer anthers. Moreover, conspicuous painting of the outer anthers of freshly opened *C. unguiculata* flowers makes captive bumblebees more likely to collect pollen on their initial visit and increases the overall proportion of pollen collecting visits. Although initial crypsis of the outer anthers is consistent with both hypotheses for heteranthery, further evidence contradicts division of labour. In the field, bee responses to anther removal change over the course of flower development. As outer anthers expose pollen and become erect, bees reduce visitation to flowers missing outer anthers, suggesting that the outer anthers become conspicuous once they are active. In the field, we regularly observe bees aggressively harvesting pollen from outer anthers once they dehisce. However, in our experiment with captive bees, forcing the undehisced outer anthers into an erect position had no effect on bee behavior. Anther movement on its own may not affect pollen collection, but instead acts in concert with dehiscence. Our captive experiment also may not adequately capture natural bee responses. For example, we had removed the conspicuous inner anthers, which may serve to draw bees to the erect outer anthers in unmanipulated flowers even after their pollen has been removed.

Finally, we test the putative benefit of pollen dosing by quantifying pollen export from *C. unguiculata* flowers that had either fully or partially dehisced outer anthers. Although dosing flowers have far less exposed pollen, they export similar quantities to stigmas during each foraging bout. Importantly, they retain pollen that could be available for future bee visits, whereas fully dehisced flowers are usually stripped bare. Thus, the slow dehiscence of the outer anthers could increase pollen export.

As in animals, male fitness in plants is typically limited by access to mates, and floral characters that promote pollen removal and delivery can evolve under intrasexual selection [38–41]. Indirect male–male competition for mates among plants in the population may be the primary selective influence on anther phenotypes in *Clarkia*. In bee-pollinated plants, if large quantities of pollen are removed during a visit, excess pollen may be lost, the bee may be stimulated to groom, and there may be little pollen remaining for future visitors who could disperse pollen to other receptive mates [18,40,42–44]. Thus, simultaneous pollen presentation may provide diminishing male fitness returns, driving selection for gradual pollen release across multiple bee visits [18] (reviewed in [20,21]). Pollen presentation theory has shed light on the evolution of pollen release schedules within and across flowers, inflorescences and pollination syndromes (e.g. [40,42,45]), yet has not been invoked to explain heteranthery. It may be that in plants like *Clarkia*, with oligolectic bee pollinators highly attuned to pollen rewards, gradual dehiscence is not sufficient to prevent pollen removal. It appears that both the colour and positional crypsis of heteranthery provide additional protection against overharvesting of pollen by bees.

We further propose that delayed anthesis of outer anthers is an inexpensive way of extending the male gain curve *sensu* Charnov [46]. *Clarkia* flowers are costly in terms of water loss during the rapidly drying Mediterranean climate spring [47]. Floral lifespan decreases substantially with increased

temperature and drought stress [48] (K.M.K. 2017, personal observation). Gradual pollen delivery may increase male fitness under favourable conditions without the investment required by an entirely new flower, yet allow the plant to quickly minimize costs by cycling faster under stressful conditions.

Our work shows conclusively that heteranthery in *Clarkia* is not explained by division of labour between pollinating and feeding anthers, and we propose pollen dosing as an alternative explanation for heteranthery in other plants. In fact, recent evidence from heterantherous *Senna* (Fabaceae) and *Adelobotrys* (Melastomataceae) shows different pollen dosing strategies between anther types [16]. In *Clarkia*, heteranthery slows pollen presentation and has the potential to increase siring success through pollen export. Flowers essentially hide available pollen with cryptic colouration, reflexed positioning and delayed dehiscence, and then gradually reveal it to bees. Darwin was right to be puzzled by heteranthery as he explored floral adaptations that promote outcrossing (reviewed in [4]). Heteranthery counterintuitively maximizes outcrossing by limiting pollen removal during bee visitation. Müller [7] insightfully noticed that the partners in bee pollination have conflicting objectives and correctly proposed that heteranthery restricts bee access to rewarding pollen. Nevertheless, here we show that with heteranthery, plants can optimize bee behavior for their own reproductive benefits in a way that is more subtle and flexible than imagined by the division of labour hypothesis. We predict that, when examined carefully, other heterantherous taxa with

specialized bee pollinators share this pollen dosing strategy. More broadly, our work highlights the importance of considering male fitness in plants.

Darwin may have reached a similar conclusion, had he lived longer. On 21 March 1881, he wrote to botanist Thiselton-Dyer of his keen interest in renewing the 'labourious experiments' he had started on heteranthery, in light of Müller's 'novel and very curious explanation', noting that he had just requested *Clarkia unguiculata* seeds from a colleague with which to experiment [49]. Given more time, Darwin would undoubtedly have noticed the anther timing difference that makes division of labour implausible.

Ethics. Field work was performed at Pinnacles National Park (permits PINN-2017-SCI-0001, PINN-2018-SCI-0003, PINN-2019-SCI-0009). Work with captive bumblebees was performed under California Department of Food and Agriculture permit 3438.

Data accessibility. All data and code are available from the Dryad Digital Repository: https://dx.doi.org/10.5061/dryad.8cz8w9gmp [50].

Authors' contributions. K.M.K. and T.J. designed the study; K.M.K., T.J., D.T. and S.A. collected the data; D.T. analysed the colour data and K.M.K. performed all other analyses; K.M.K. wrote the manuscript and all the authors edited the manuscript.

Competing interests. We declare we have no competing interests.

Funding. This work was funded by a UCSC COR grant to KMK, a Kenneth S. Norris Center student award to D.T., and a CC-RISE award to T.J. and E. McCoy.

Acknowledgements. We thank J. Velzy and S. Childress for expert plant care and facilitation of bee trials, Pinnacles National Park and P. Johnson for field support, E. McCoy for counting pollen on bees, and C. J. van der Kooi for advice with colour analysis.

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
