## [Reviewer comments · Proceedings of the Royal Society B: Biological Sciences]

Review History

RSPB-2020-1454.R0 (Original submission)

Review form: Reviewer 1 (James Thomson)

Recommendation

Accept with minor revision (please list in comments)

Scientific importance: Is the manuscript an original and important contribution to its field?

Excellent

General interest: Is the paper of sufficient general interest?

Good

Quality of the paper: Is the overall quality of the paper suitable?

Excellent

Is the length of the paper justified?

Yes

Should the paper be seen by a specialist statistical reviewer?

No

Do you have any concerns about statistical analyses in this paper? If so, please specify them explicitly in your report.

Yes

It is a condition of publication that authors make their supporting data, code and materials available - either as supplementary material or hosted in an external repository. Please rate, if applicable, the supporting data on the following criteria.

Is it accessible?

N/A

Is it clear?

N/A

Is it adequate?

N/A

Do you have any ethical concerns with this paper?

No

Comments to the Author

I have attached a Word file. I see no point in anonymity. (See Appendix A)

Review form: Reviewer 2

Recommendation

Reject – article is scientifically unsound

Scientific importance: Is the manuscript an original and important contribution to its field?

Acceptable

General interest: Is the paper of sufficient general interest?

Good

Quality of the paper: Is the overall quality of the paper suitable?

Acceptable

Is the length of the paper justified?

Yes

Should the paper be seen by a specialist statistical reviewer?

No

Do you have any concerns about statistical analyses in this paper? If so, please specify them explicitly in your report.

Yes

It is a condition of publication that authors make their supporting data, code and materials available - either as supplementary material or hosted in an external repository. Please rate, if applicable, the supporting data on the following criteria.

Is it accessible?

N/A

Is it clear?

N/A

Is it adequate?

N/A

Do you have any ethical concerns with this paper?

No

Comments to the Author

In this study, Kay and colleagues investigate the evolution and function of differently coloured anthers in the *Clarkia*. The authors propose a hypothesis to explain the different colouration of anthers in these species, namely, selection for gradual pollen presentation. Below I provide some major and minor suggestions.

Major suggestions:

- 1) The definition of heteranthery (eg., abstract and line 37) used here needs to be refined (what qualifies as “types”? is a difference in colour sufficient?). Two morphologically different sets of stamens is not necessarily heteranthery, as commonly understood by plant evolutionary biologists. For example, stamens with different filament lengths in Brassicaceae or Scrophulariaceae are not considered heterantherous. This may seem like a small detail, but one could argue that it lays at the foundation of the thesis presented in this manuscript. If heteranthery is defined as any flower in which anthers are not morphologically the same, then it is hardly surprising that no single hypothesis can explain the evolution of what are certainly not evolutionary convergent structures.
- 2) The finding that heteranthery evolved once in *Clarkia* (line 329) raises some interesting questions about whether this is a good system for studying the evolution of heteranthery. With a single evolutionary transition it is probably not possible to meaningfully conduct statistical tests of character correlations between heteranthery and other features. Would it be possible to please estimate and provide in the text the number of evolutionary losses of heteranthery inferred in this study? From Figure 1 one can see a single gain and two losses of heteranthery. With no phylogenetic replication, it is possible to imagine that anther colour dimorphism in the flowers of some *Clarkia* do not necessarily serve the same function that caused the evolution of dimorphism in the first place.
- 3) It is not clear how the new hypothesis proposed here (gradual pollen release) can explain the evolution of heteranthery. Gradual pollen release can be achieved through a variety of ways, and even heterantherous species show other strategies for gradual pollen release. Why has heteranthery evolved in species which already possess gradual pollen release? In which conditions would heteranthery evolve to dose pollen instead of other (simpler?) strategies such as gradual anther maturation, etc.
- 4) In line 52-53 it is said that “it would be naïve to assume that bees, which have long coevolved to forage efficiently, will often overlook available pollen in inconspicuous anthers.” This may seem like an unfortunate wording as it implies that flowers cannot “fool” pollinators. It could be argued that there are plenty of examples of floral crypsis and deceptive pollination that show that pollinators indeed can be manipulated to visit flowers even when this is not necessarily the most efficient behaviour for the floral visitor.
- 5) In line 101-104 the authors state that to test their new hypothesis they compare pollen export in plants with and without gradual dehiscence. However, one could argue that this experiment cannot possibly be used as a way to refute the division of labour hypothesis since gradual pollen presentation may not be a strategy that is mutually exclusive with the division of labour. Gradual pollen dispensing is likely favoured in most species of plants that receive a sufficient number of floral visits whether they are heterantherous or not.
- 6) A potential experimental problem is that pollen in the scopae was used to assess pollen export. Pollen in the scopae is removed from the pollination process and in many cases,

particularly in some heterantherous species, the pollen transferred to stigmas is the one that is more likely to escape from bee packing in the scopae. The spatial arrangement of feeding and pollinating stamens might make pollinating anthers more likely to place pollen in parts of the bee's body which are harder to groom (safe sites) and thus the ration of pollen from the two anther types in the scopae is not a good reflection of pollen export. In fact, I would interpret the finding (lines 360-362) of less than expected purple, outer ("pollinating stamen") pollen in the scopae as CONSISTENT with the division of labour hypothesis.

Minor suggestions:

- 7) Line 25. Suggested change: "We find no support for division of labour in Clarkia, but multifarious..."
- 8) 157-158. Please clarify. Simultaneous presentation would refer to the two anther types offering some pollen at the same time. But this does not preclude gradual presentation of pollen (eg. gradual presentation of pollen as the flower ages)
- 9) Line 166. Only one flower was studied per species? What is the variation among flowers of the same plant/species?
- 10) Line 177 and earlier. Please define colour heteranthery.
- 11) Lines 180-182. Could you provide more details of how the model was parametrised?
- 12) Line 183. Was this a different test (combination of traits) than above?
- 13) Line 184-186. Can you please provide more detail of how this model was constructed?
- 14) Does the pan trap with soapy water remove pollen from bees? This might be likely especially if bees stayed in the pans for the full collecting period (4 hours)

Review form: Reviewer 3

Recommendation

Major revision is needed (please make suggestions in comments)

Scientific importance: Is the manuscript an original and important contribution to its field?

Excellent

General interest: Is the paper of sufficient general interest?

Good

Quality of the paper: Is the overall quality of the paper suitable?

Good

Is the length of the paper justified?

Yes

Should the paper be seen by a specialist statistical reviewer?

No

Do you have any concerns about statistical analyses in this paper? If so, please specify them explicitly in your report.

Yes

It is a condition of publication that authors make their supporting data, code and materials available - either as supplementary material or hosted in an external repository. Please rate, if applicable, the supporting data on the following criteria.

Is it accessible?

Yes

Is it clear?

Yes

Is it adequate?

Yes

Do you have any ethical concerns with this paper?

Yes

Comments to the Author

In this manuscript, Kay et al. studied *Clarkia* species to propose and validate a new hypothesis to investigate the function of heteranthery, an outcross pollination hypothesis, in contrast to the division of labour hypothesis. I am impressed with the amount of work conducted, and I enjoyed the multiple experiments and observations gathered to show that the two types of anthers seem to provide similar functions (at least feeding bees). I am not an expert in heteranthery, but the division of labour hypothesis states that flowers with two types of anthers play a different role, one type for pollination, the other to feed bees, and pollinating anthers export more pollen than feeding anthers. In this study, the combination of observational/correlative data and experimental manipulations shows that heteranthery and associated traits evolved with bee pollination, and that the anthers classified as pollinating anthers are used by bees to collect pollen. They also show that anther movement and dehiscence functions to gradually deliver pollen, as opposed to full on presentation, in turn securing more visits and more pollen export in subsequent visits.

I enjoyed the question-approach of the authors to address specific functional hypotheses. However, the main flaw of the manuscript is the lack of clarity in the motivation of the study as exposed in the introduction, and the fact that the narrative used by the authors sounds “defensive” and focused on “our results don’t agree with the division of labour hypothesis but with the out-cross pollination hypothesis”. At present, it is necessary to read the introduction and the methods, and a lot of reading between the lines to gain a full picture of the two hypotheses and what the authors want to do: do you want to present a new way to understand how heteranthery evolved, or do you want to reject the division of labour hypothesis?. If the authors want to follow the narrative of conducting experiments and presenting data to see if it validates one hypothesis or the other, I suggest to present the two hypotheses at the introduction, and move the bits in the M&M that sound like “oh and by the way, under this hypothesis the expectation is...”. I have added my comments in the text where this happened. So it is important to present the division of labour and outcross-pollination hypotheses in the introduction, and hence create specific aims (already there in the different questions presented in the M&M) in relation to specific components of the hypotheses or tests to validate one or the other. What I gathered is that the authors want to demonstrate that heteranthery in *Clarkia* evolved in the context of the promotion of cross-pollination, as opposed to division of labour. Focus your narrative on that, and then you will be able to create a stronger argument for an alternative hypothesis to the division of labour.

After reading the manuscript, I was left with mixed feelings about one hypothesis or the other. What is unclear to me is why two types of anthers evolved. Division of labour is set under the context of male to male competition and intrafloral male competition. Intrafloral male to male competition doesn’t make much sense to me because it doesn’t benefit the total male fitness that a single flower can achieve. Instead, a strategy that increases pollen export and siring success would be selected (so that male gametes of a flower don’t compete, or compete less). This is where the advertisement role of inner anthers is so interesting. The anther removal experiment is really inspiring to learn how much bees rely on those anthers to perceive and visit flowers, and whether they function as advertisement so that bees learn where to go later in the afternoon, and

repeatedly as anthers open and deliver pollen. Note that plants where the division of labour has been tested also release pollen gradually. Having a set of anthers that increases visitation later in the day is a good way to increase pollen delivery and individual (flower level) male fitness, instead of intrafloral male to male competition. It would be interesting to know if the fertility of pollen from both anther types is the same, or whether inner anthers are less fertile. Then you would have a new twist in the division of labour hypothesis to promote cross-pollination.

In appendix S1, the AIC values of the models (heteranthery depends on pollination -I would say correlated with, or evolved after-) is similar (ca. 2 units larger) that a model with characters are independent and pollination depends on heteranthery...so even the first model shows the lowest AIC, it is still close to the AICs displayed by other models. I am not fully convinced that correlated evolution tests are helping here to demonstrate what are the most likely evolutionary events. It is clearer in the tests of correlated evolution between anther movement and colour of anthers. Also, use a narrative that infers correlation and not causation.

I have added specific comments in the text.

Decision letter (RSPB-2020-1454.R0)

24-Aug-2020

Dear Dr Kay:

I am writing to inform you that your manuscript RSPB-2020-1454 entitled "Darwin's vexing contrivance: a new hypothesis for why some flowers have two kinds of anthers" has, in its current form, been rejected for publication in Proceedings B.

This action has been taken on the advice of referees, who have recommended that substantial revisions are necessary. With this in mind we would be happy to consider a resubmission, provided the comments of the referees are fully addressed. However please note that this is not a provisional acceptance.

To upload a resubmitted manuscript, log into <http://mc.manuscriptcentral.com/prsb> and enter your Author Centre, where you will find your manuscript title listed under "Manuscripts with

Decisions." Under "Actions," click on "Create a Resubmission." Please be sure to indicate in your cover letter that it is a resubmission, and supply the previous reference number.

Sincerely,
Dr Locke Rowe
mailto: proceedingsb@royalsociety.org

Associate Editor
Board Member: 1
Comments to Author:

I have received three reviews of your manuscript entitled "Darwin's vexing contrivance: a new hypothesis for why some flowers have two kinds of anthers". The recommendations made by the first two reviews were divergent and I requested a third review. Going out for a third review delayed the process, and I apologize for the extended wait time.

All referees found the topic to be interesting, and referee 1 was strongly positive about the manuscript. Referees 1 and 3 both had positive comments about the ambitious, multifaceted approach taken to understand the evolution and function of heteranthery in *Clarkia*.

However, the reports from referees 2 and 3 were quite critical and referee 2 in particular has flagged issues that raise questions about the overall objectives of the study. In particular, referee 2 points out that the division of labour and pollen dosing hypotheses are not mutually exclusive. This point is also made by referee 1. I agree with this comment. But, if we accept that pollen dosing and division of labour can operate simultaneously, there would seem to be a problem for the manuscript, which sets up the two hypotheses as "competing" with each other and that the series of analyses presented in the manuscript can distinguish the "alternative" hypotheses. This point was further driven home by referee 2, who points out that pollen dosing does not require heteranthery to operate - many non-heterantherous flowers engage in pollen dosing. Moreover, if we again accept that the two hypotheses are not mutually exclusive (and/or that the definition regarding the presentation of pollen under division of labour is too narrow - a point made by referee 1), at least one of the experiments does not seem to enable a test of the two alternative hypotheses. The pollen dosing experiment described on Pgs. 11 & 12 seems to be a test of the function of pollen dosing (gradual versus simultaneous pollen release) in promoting pollen export, not a test of pollen dosing versus division of labour.

Referee 2 has also flagged concerns with the analyses (comment #2) and referees 2 and 3 questioned different aspects of data interpretation (referee 2: comment #6, referee 3: final comment in the comments to authors section). I found these points to be persuasive, further weakening the conclusions of the study.

On balance, this manuscript has several strengths, the most notable of which is the multifaceted approach to the study of a topic of interest to a broad audience. However, it has several weaknesses that raise questions about the general framework of the study and its conclusions. Addressing these weaknesses, particularly those flagged by referee 2, would appear to require a major reworking of the manuscript.

Reviewer(s)' Comments to Author:

Referee: 1
Comments to the Author(s)
I have attached a Word file. I see no point in anonymity.

Referee: 2
Comments to the Author(s)
In this study, Kay and colleagues investigate the evolution and function of differently coloured anthers in the *Clarkia*. The authors propose a hypothesis to explain the different colouration of

anthers in these species, namely, selection for gradual pollen presentation. Below I provide some major and minor suggestions.

Major suggestions:

- 1) The definition of heteranthery (eg., abstract and line 37) used here needs to be refined (what qualifies as “types”? is a difference in colour sufficient?). Two morphologically different sets of stamens is not necessarily heteranthery, as commonly understood by plant evolutionary biologists. For example, stamens with different filament lengths in Brassicaceae or Scrophulariaceae are not considered heterantherous. This may seem like a small detail, but one could argue that it lays at the foundation of the thesis presented in this manuscript. If heteranthery is defined as any flower in which anthers are not morphologically the same, then it is hardly surprising that no single hypothesis can explain the evolution of what are certainly not evolutionary convergent structures.
- 2) The finding that heteranthery evolved once in *Clarkia* (line 329) raises some interesting questions about whether this is a good system for studying the evolution of heteranthery. With a single evolutionary transition it is probably not possible to meaningfully conduct statistical tests of character correlations between heteranthery and other features. Would it be possible to please estimate and provide in the text the number of evolutionary losses of heteranthery inferred in this study? From Figure 1 one can see a single gain and two losses of heteranthery. With no phylogenetic replication, it is possible to imagine that anther colour dimorphism in the flowers of some *Clarkia* do not necessarily serve the same function that caused the evolution of dimorphism in the first place.
- 3) It is not clear how the new hypothesis proposed here (gradual pollen release) can explain the evolution of heteranthery. Gradual pollen release can be achieved through a variety of ways, and even heterantherous species show other strategies for gradual pollen release. Why has heteranthery evolved in species which already possess gradual pollen release? In which conditions would heteranthery evolve to dose pollen instead of other (simpler?) strategies such as gradual anther maturation, etc.
- 4) In line 52-53 it is said that “it would be naïve to assume that bees, which have long coevolved to forage efficiently, will often overlook available pollen in inconspicuous anthers.” This may seem like an unfortunate wording as it implies that flowers cannot “fool” pollinators. It could be argued that there are plenty of examples of floral crypsis and deceptive pollination that show that pollinators indeed can be manipulated to visit flowers even when this is not necessarily the most efficient behaviour for the floral visitor.
- 5) In line 101-104 the authors state that to test their new hypothesis they compare pollen export in plants with and without gradual dehiscence. However, one could argue that this experiment cannot possibly be used as a way to refute the division of labour hypothesis since gradual pollen presentation may not be a strategy that is mutually exclusive with the division of labour. Gradual pollen dispensing is likely favoured in most species of plants that receive a sufficient number of floral visits whether they are heterantherous or not.
- 6) A potential experimental problem is that pollen in the scopae was used to assess pollen export. Pollen in the scopae is removed from the pollination process and in many cases, particularly in some heterantherous species, the pollen transferred to stigmas is the one that is more likely to escape from bee packing in the scopae. The spatial arrangement of feeding and pollinating stamens might make pollinating anthers more likely to place pollen in parts of the bee’s body which are harder to groom (safe sites) and thus the ration of pollen from the two anther types in the scopae is not a good reflection of pollen export. In fact, I would interpret the finding (lines 360-362) of less than expected purple, outer (“pollinating stamen”) pollen in the scopae as CONSISTENT with the division of labour hypothesis.

Minor suggestions:

- 7) Line 25. Suggested change: “We find no support for division of labour in *Clarkia*, but multifarious...”

- 8) 157-158. Please clarify. Simultaneous presentation would refer to the two anther types offering some pollen at the same time. But this does not preclude gradual presentation of pollen (eg. gradual presentation of pollen as the flower ages)
- 9) Line 166. Only one flower was studied per species? What is the variation among flowers of the same plant/species?
- 10) Line 177 and earlier. Please define colour heteranthery.
- 11) Lines 180-182. Could you provide more details of how the model was parametrised?
- 12) Line 183. Was this a different test (combination of traits) than above?
- 13) Line 184-186. Can you please provide more detail of how this model was constructed?
- 14) Does the pan trap with soapy water remove pollen from bees? This might be likely especially if bees stayed in the pans for the full collecting period (4 hours)

Referee: 3

Comments to the Author(s)

In this manuscript, Kay et al. studied *Clarkia* species to propose and validate a new hypothesis to investigate the function of heteranthery, an outcross pollination hypothesis, in contrast to the division of labour hypothesis. I am impressed with the amount of work conducted, and I enjoyed the multiple experiments and observations gathered to show that the two types of anthers seem to provide similar functions (at least feeding bees). I am not an expert in heteranthery, but the division of labour hypothesis states that flowers with two types of anthers play a different role, one type for pollination, the other to feed bees, and pollinating anthers export more pollen than feeding anthers. In this study, the combination of observational/correlative data and experimental manipulations shows that heteranthery and associated traits evolved with bee pollination, and that the anthers classified as pollinating anthers are used by bees to collect pollen. They also show that anther movement and dehiscence functions to gradually deliver pollen, as opposed to full on presentation, in turn securing more visits and more pollen export in subsequent visits.

I enjoyed the question-approach of the authors to address specific functional hypotheses. However, the main flaw of the manuscript is the lack of clarity in the motivation of the study as exposed in the introduction, and the fact that the narrative used by the authors sounds “defensive” and focused on “our results don’t agree with the division of labour hypothesis but with the out-cross pollination hypothesis”. At present, it is necessary to read the introduction and the methods, and a lot of reading between the lines to gain a full picture of the two hypotheses and what the authors want to do: do you want to present a new way to understand how heteranthery evolved, or do you want to reject the division of labour hypothesis?. If the authors want to follow the narrative of conducting experiments and presenting data to see if it validates one hypothesis or the other, I suggest to present the two hypotheses at the introduction, and move the bits in the M&M that sound like “oh and by the way, under this hypothesis the expectation is...”. I have added my comments in the text where this happened. So it is important to present the division of labour and outcross-pollination hypotheses in the introduction, and hence create specific aims (already there in the different questions presented in the M&M) in relation to specific components of the hypotheses or tests to validate one or the other. What I gathered is that the authors want to demonstrate that heteranthery in *Clarkia* evolved in the context of the promotion of cross-pollination, as opposed to division of labour. Focus your narrative on that, and then you will be able to create a stronger argument for an alternative hypothesis to the division of labour.

After reading the manuscript, I was left with mixed feelings about one hypothesis or the other. What is unclear to me is why two types of anthers evolved. Division of labour is set under the context of male to male competition and intrafloral male competition. Intrafloral male to male competition doesn’t make much sense to me because it doesn’t benefit the total male fitness that a single flower can achieve. Instead, a strategy that increases pollen export and siring success would be selected (so that male gametes of a flower don’t compete, or compete less). This is where the advertisement role of inner anthers is so interesting. The anther removal experiment is

really inspiring to learn how much bees rely on those anthers to perceive and visit flowers, and whether they function as advertisement so that bees learn where to go later in the afternoon, and repeatedly as anthers open and deliver pollen. Note that plants where the division of labour has been tested also release pollen gradually. Having a set of anthers that increases visitation later in the day is a good way to increase pollen delivery and individual (flower level) male fitness, instead of intrafloral male to male competition. It would be interesting to know if the fertility of pollen from both anther types is the same, or whether inner anthers are less fertile. Then you would have a new twist in the division of labour hypothesis to promote cross-pollination.

In appendix S1, the AIC values of the models (heteranthery depends on pollination -I would say correlated with, or evolved after-) is similar (ca. 2 units larger) that a model with characters are independent and pollination depends on heteranthery...so even the first model shows the lowest AIC, it is still close to the AICs displayed by other models. I am not fully convinced that correlated evolution tests are helping here to demonstrate what are the most likely evolutionary events. It is clearer in the tests of correlated evolution between anther movement and colour of anthers. Also, use a narrative that infers correlation and not causation.

I have added specific comments in the text.

Author's Response to Decision Letter for (RSPB-2020-1454.R0)

See Appendix B.

RSPB-2020-2593.R0

Review form: Reviewer 1

Recommendation

Accept as is

Scientific importance: Is the manuscript an original and important contribution to its field?

Excellent

General interest: Is the paper of sufficient general interest?

Good

Quality of the paper: Is the overall quality of the paper suitable?

Excellent

Is the length of the paper justified?

Yes

Should the paper be seen by a specialist statistical reviewer?

No

Do you have any concerns about statistical analyses in this paper? If so, please specify them explicitly in your report.

No

It is a condition of publication that authors make their supporting data, code and materials available - either as supplementary material or hosted in an external repository. Please rate, if applicable, the supporting data on the following criteria.

Is it accessible?

N/A

Is it clear?

N/A

Is it adequate?

N/A

Do you have any ethical concerns with this paper?

No

Comments to the Author

I considered the first version a valuable piece of work, but hoped to see a tightening of definitions and more explanation of how observations bore on the question. The revision has accomplished that.

Review form: Reviewer 3

Recommendation

Accept with minor revision (please list in comments)

Scientific importance: Is the manuscript an original and important contribution to its field?

Excellent

General interest: Is the paper of sufficient general interest?

Good

Quality of the paper: Is the overall quality of the paper suitable?

Good

Is the length of the paper justified?

Yes

Should the paper be seen by a specialist statistical reviewer?

No

Do you have any concerns about statistical analyses in this paper? If so, please specify them explicitly in your report.

No

It is a condition of publication that authors make their supporting data, code and materials available - either as supplementary material or hosted in an external repository. Please rate, if applicable, the supporting data on the following criteria.

Is it accessible?

Yes

Is it clear?

Yes

Is it adequate?

Yes

Do you have any ethical concerns with this paper?

No

Comments to the Author

This is a new version of a manuscript that I reviewed previously. The authors have made great effort to improve the text and to clarify the motivation of the study. I think that the new version is much improved, and it will make a strong contribution to the Proceedings of the Royal Society series B by providing new evidence to the origin and function of heteranthery. I am somewhat concerned about a few aspects of the manuscript, which I hope the authors can clarify.

A key aspect to test the hypothesis of division of labour vs. pollen dosing is to provide good estimates of male fitness. But the male fitness proxies are inferred from bee behaviour and pollen collection, and through pollen deposition on the stigmas. With regards the later, this is true since what limits male fitness is access to partners, and in this manuscript is assessed by counting pollen load on the stigmas. But the authors showed that the two types of anthers don't seem to differ (too strongly) on the amount of pollen deposited on stigmas (at least in one of the experiments). Access to partners can also be quantified by the % flowers with pollen from one or the other anther type. And indeed that could be (potentially) a better proxy of male fitness. The amount of pollen on the stigmatic surface depends on the total stigmatic area, which could (also potentially) express variation among flowers (this is in relation to line 856 and line 1002)

The experimental manipulation with *Bombus* is extremely smart and clearly laborious. But I wonder to what degree the results obtained with *Bombus* are comparable to those retrieved using native bees. I assume that *Bombus* and *Hesperapis regularis* are different in size, behaviour etc so that I am not entirely sure if the magnitude of the results obtained in the experiments involving two different bee species are comparable. Probably a qualitative comparison is possible, but it would be, at least good to acknowledge this. After all, heteranthery has evolved (and functions) in the context of local pollinators.

Line 756: I wonder if some of the patterns identified can be explained because native bees learn to search based on their experience. So for example, export of pollen from outer anthers could increase if bees had a positive experience in the same flower with inner anthers, and they could simply learn "where the good flowers for pollen are". Is there a correlation between pollen production between inner and outer anthers? Although this calls for a quantitative (not qualitative) comparison.

Line 813-819: This is precisely my point mentioned above. You might detect a signal if you estimate the number of flowers with pollen from each anther type, instead of quantifying pollen on the stigmas. The former is also a way to quantify access to partners (access to partners in terms of different individual flowers as well as ovules in one flower).

End of comments

Decision letter (RSPB-2020-2593.R0)

19-Nov-2020

Dear Dr Kay

I am pleased to inform you that your manuscript RSPB-2020-2593 entitled "Darwin's vexing contrivance: a new hypothesis for why some flowers have two kinds of anthers" has been accepted for publication in Proceedings B.

The referee(s) have recommended publication, but also suggest some minor revisions to your manuscript. Therefore, I invite you to respond to the referee(s)' comments and revise your manuscript. Because the schedule for publication is very tight, it is a condition of publication that you submit the revised version of your manuscript within 7 days. If you do not think you will be able to meet this date please let us know.

In order to ensure effective and robust dissemination and appropriate credit to authors the dataset(s) used should be fully cited. To ensure archived data are available to readers, authors

should include a 'data accessibility' section immediately after the acknowledgements section. This should list the database and accession number for all data from the article that has been made publicly available, for instance:

[http://datadryad.org/submit?journalID=RSPB&manu=\(Document not available\)](http://datadryad.org/submit?journalID=RSPB&manu=(Document%20not%20available)) which will take you to your unique entry in the Dryad repository. If you have already submitted your data to dryad you can make any necessary revisions to your dataset by following the above link. Please see <https://royalsociety.org/journals/ethics-policies/data-sharing-mining/> for more details.

Sincerely,

Dr Locke Rowe

Associate Editor

Board Member

Comments to Author:

Thank you for resubmitting your manuscript to Proceedings B. Both reviewers had assessed the previous version of the manuscript and both report that the revised manuscript (and rebuttal) addresses the main comments raised in that round of reviews. I agree with their assessments. Reviewer 1 continues to be very positive about this study and is not suggesting any revision. Reviewer 2 makes several comments in relation to the results and discussion. I have only one minor comment.

L219 - For the study described as "Are outer anthers cryptic, and do they become conspicuous to bees as they release pollen?" indicate the number of flowers sampled per species, and the stage at which flowers were sampled (before, or during pollen dehiscence of the inner and outer whorls of anthers).

Reviewer(s)' Comments to Author:

Referee: 1

Comments to the Author(s).

I considered the first version a valuable piece of work, but hoped to see a tightening of definitions and more explanation of how observations bore on the question. The revision has accomplished that.

Referee: 3

Comments to the Author(s).

This is a new version of a manuscript that I reviewed previously. The authors have made great effort to improve the text and to clarify the motivation of the study. I think that the new version is much improved, and it will make a strong contribution to the Proceedings of the Royal Society series B by providing new evidence to the origin and function of heteranthery. I am somewhat concerned about a few aspects of the manuscript, which I hope the authors can clarify.

A key aspect to test the hypothesis of division of labour vs. pollen dosing is to provide good estimates of male fitness. But the male fitness proxies are inferred from bee behaviour and pollen collection, and through pollen deposition on the stigmas. With regards the later, this is true since what limits male fitness is access to partners, and in this manuscript is assessed by counting pollen load on the stigmas. But the authors showed that the two types of anthers don't seem to differ (too strongly) on the amount of pollen deposited on stigmas (at least in one of the experiments). Access to partners can also be quantified by the % flowers with pollen from one or the other anther type. And indeed that could be (potentially) a better proxy of male fitness. The amount of pollen on the stigmatic surface depends on the total stigmatic area, which could (also potentially) express variation among flowers (this is in relation to line 856 and line 1002)

The experimental manipulation with *Bombus* is extremely smart and clearly laborious. But I wonder to what degree the results obtained with *Bombus* are comparable to those retrieved using native bees. I assume that *Bombus* and *Hesperapis regularis* are different in size, behaviour etc so that I am not entirely sure if the magnitude of the results obtained in the experiments involving two different bee species are comparable. Probably a qualitative comparison is possible, but it would be, at least good to acknowledge this. After all, heteranthery has evolved (and functions) in the context of local pollinators.

Line 756: I wonder if some of the patterns identified can be explained because native bees learn to search based on their experience. So for example, export of pollen from outer anthers could increase if bees had a positive experience in the same flower with inner anthers, and they could simply learn "where the good flowers for pollen are". Is there a correlation between pollen production between inner and outer anthers? Although this calls for a quantitative (not qualitative) comparison.

Line 813-819: This is precisely my point mentioned above. You might detect a signal if you estimate the number of flowers with pollen from each anther type, instead of quantifying pollen on the stigmas. The former is also a way to quantify access to partners (access to partners in terms of different individual flowers as well as ovules in one flower).

End of comments

Author's Response to Decision Letter for (RSPB-2020-2593.R0)

See Appendix C.

Decision letter (RSPB-2020-2593.R1)

26-Nov-2020

Dear Dr Kay

I am pleased to inform you that your manuscript entitled "Darwin's vexing contrivance: a new hypothesis for why some flowers have two kinds of anthers" has been accepted for publication in Proceedings B.

Open Access

Paper charges

Sincerely,

Appendix A

This paper seeks to compare two alternative (and ostensibly competing, line 14) explanations for the evolution of heteranthy: division of labor (DOL) versus pollen dosing (PD). As it happens, I have a foot in each of these camps, in that I'm a minor coauthor on a paper that is thought to provide one of the more convincing arguments for DOL (Vallejo-Marin et al. 2009), but have also been involved in promoting PD as a widespread and potent selective force in floral evolution (Harder and Thomson 1989, etc.). Therefore, it seems simplest to waive anonymity.

This is an exemplary paper that offers a choice of explanations for an intriguing phenomenon, identifies an illustrative plant genus, and explores the evidence in a multi-pronged way that I would characterize as "using every blade on the Swiss Army knife." I find this an appealing and powerful approach. The writing is clear and efficient, and could be published virtually "as is" without embarrassment to the authors or the journal. Nevertheless, I do have some suggestions for some further development.

Admirably, the introduction gets right to the point, but I feel that the contrast between DOL and GRP needs more development. Specifically, the two hypotheses need to be stated in terms of the specific points of differentiation that will be used to distinguish them. At a couple of places later in the MS (e.g., lines 247-8) Kay et al. refer to "predictions of our pollen dosing hypothesis" in ways that left me a little surprised or unprepared. I did not think that those predictions—and their supporting rationale—had been adequately stated in the Intro. Therefore, I had to do some conceptual backtracking to satisfy myself about the reasoning behind the predictions. This is the wrong place in the paper for a reader to be working this out. I would find it more comfortable to have gotten a more explicit account, perhaps even a list of predictions, in the Introduction. The current Introduction seems to depend on the reader to infer, for example, the critical defining characteristics of DOL from little more than the name itself. I, for one, needed more hand-holding to fully understand the authors' arguments.

The most important example of this shows up in line 66, where DOL between feeding and pollinating anthers is stipulated to occur "*during the same bee visit.*" If I infer correctly, that seemingly innocuous little phrase about timing is *essential* to their operational definition of DOL. I see it as a restrictive definition that gives rise to a "narrow-sense" view of DOL. One could imagine the words "division of labor" being interpreted more broadly. For example, if rewards gleaned from earlier-dehiscing anthers served to entrain return visits by bees that would move pollen from later-opening anthers, I could interpret that as a division of labor. More broadly, I think that I could view *any* differences in the timing of anther function as DOL in the sense of the English words, but Kay et al. would reject such interpretations. If these definitional matters can be attended to in the Introduction, the later portions of the paper will read more smoothly because readers won't be waylaid by niggles.

Another bigish point is that I think that one could entertain a third hypothesis, that the two whorls of stamens might be optimized for working with different species of pollinators, i.e., heteranthy might be a way of producing a generalist floral morphology that can be well served by visitors of different sizes and proclivities. Of course, this too could be considered a variety of DOL, but I didn't think it's the kind of DOL that Kay et al. are stipulating.

I'll try to amplify this point by going through the paper and flagging various items by line number (along with any suggestions for minor fixups).

14. Must these hypotheses be *competing*? What prevents both from operating simultaneously?

43. The word “stumped” is evocative and effective, but might be too colloquial for non-native speakers?

65 and elsewhere. I infer that these anthers dehisce gradually along their length, thereby allowing gradual pollen presentation within an anther (hence measurements in mm at line 438). It would be good to clarify explain this more (unless I missed it).

66. Here’s the key phrase “during the same bee visit.” Emphasize?

68. Important but vague. Consider explaining “worse performance” in terms of what was actually measured.

88. *Must* DOL predict simultaneous pollen release? Why? References? (See general comment above.) Might not DOL extend to the first whorl serving to attract and entrain site-faithful visitors, and the second whorl to placing pollen optimally on them when they come back?

93. Good: “both types exported.” This is exactly the way to clarify the different propositions. Lines 92-97 are excellent, clear statements. Still, they could be bolstered by explaining their rationales to pacify skeptics who aren’t convinced by the unadorned assertions.

121. In what sense is “primarily” used and measured here? Purely by frequency of visits, or do other considerations factor in?

125. Are “greenhouse and live” two categories of study, or a compound description of a single type? I think the latter, but the phrasing could be improved by rearrangement.

138. Better to change “while” to “although” here (and everywhere else where simultaneity is not the intended meaning).

147. It’s a picky point, but I would also expect it with pollination by pollen-collecting masarid wasps...

171. Define “actively dehiscing.” Gradual unzipping?

204. Did you make observations to ensure that drowning in bowls doesn’t disrupt pollen on bee bodies? And that it doesn’t affect pollen color? Based on my experience with a pollen dichromism in *Erythronium*, I also worry that color might be diluted or vanish entirely when grains germinate on stigmas. (Thomson, J. D. 1989. Germination schedules of pollen grains: implications for pollen selection. *Evolution* 43:220-223.)

220-221. Here’s one of the asserted predictions about the competing hypotheses that made me uneasy.

230 and surrounding lines. Obscure technical jargon, e.g., “boxcar width,” WTF? Consider relegating these details to an online supplement.

238. Please explain the biological significance and behavioral implications of green contrast (here, not in a supplement).

246. Bees “only” notice the inner whorl?—this sounds too strong. Wouldn’t a partial preference suffice?

247. In my conception of pollen dosing, it would work perfectly well without any recognition by the bees—as it does in the many species that dose but are not heterantherous. I must be missing something.

259. Capitalize Poisson, I think.

279. “Free access” = what exactly? Ok, I see you treat this below. Presumably these were worker bees. (Actually, contrasting workers with males (non-pollen-seeking) would be an interesting idea.) Their motivation to collect pollen may depend pretty strongly on the number and hunger level of brood in the nest. If she is desperate for pollen, even a single worker could strip all of the available pollen from 15 flowers. Did you pay attention to the size of the bees? This can vary a lot in captive bees, depending on how they have been fed.

310-313. Again, the implications of “length of exposed pollen” would be easier to understand with more explanation earlier in the paper. This is a clever experiment, but I wonder if the contrast between just-starting-to dehisce anthers and fully-dehisced anthers is really adequate to isolate a pure dosing effect. I would worry about confounding with pollen age and desiccation, which is likely to affect stickiness, electrostatics, and transfer characteristics.

317. I am surprised that you found it feasible to weigh corbicular loads from such short foraging bouts. I would never have thought to try this. Even if you could get the pollen off as intact pellets, I would worry about variance induced by bee size, motivation level, etc. If you can give some extra detail about feasibility/repeatability, others might be encouraged to try this. Should you do more in the future, looking separately at both sides of the bee would give some insight into the repeatability question.

343. In Figure 1: So, what genius taxonomist decided to name one of the no-heteranthery species *C. heterandra*? Might there be a story there?

352. Figure 2. Very nice. I see that at some point in the text that you introduce the term “color heteranthery.” It might be better to do that earlier, and it would be good to specify whether that’s what you mean by heteranthery in this table.

360 onward. By contrasting the color ratios in the scopae to what’s on the stigmas, you seem to be implying that the scopal loads are entirely “feeding” pollen. This would be true for apid bees with corbiculae, but it’s not as clear-cut for bees with looser scopal loads. The grains in the scopae are probably still available for stigmatic deposit. This does not invalidate the important of finding a difference in the ratios of the two pools of pollen, but it might be worth pointing out, especially because the paper includes results from both corbiculate and non-corbiculate bees. There’s some material on the differences here: Parker, A.J., J.L. Tran, J.L. Ison, J.D.K. Bai, A.E. Weis, and J.D. Thomson. 2015. Pollen packing affects the function of pollen on corbiculate bees but not non-corbiculate bees. *Arthropod-Plant Interactions* 9:197-203.

369, Fig. 3. I like the paired panels in this figure.

370. In relation to the comment on line 360, perhaps the increase in purple pollen with total cuints could be driven by visits that happened to result in particularly intimate contact with scopal pollen. It would be good to know a little more mechanical natural history about how bee size, bee behavior, and floral morphology interact to produce contacts and transfer.

379 and elsewhere. There’s disagreement between Figure numbers in text and captions for 5 and 6. Looks like a former Fig. 4 was deleted?

388 and following. Again, I am struck by the possibility that early rewards might influence later visitation rates (which might arguably be considered a kind of DOL). It would also be helpful to know more about how many flowers these plants produce, and over how many days? With bumble bees, at least, there is evidence that early-stage rewards by certain plants can confer higher visitation to later stages. I'm not up to date on the literature on such spatial-memory-holdover effects, but here are some older papers that I consider relevant. Bees that don't trapline might not show such effects.

Cartar RV (2004) Resource tracking by bumble bees: responses to plant-level differences in quality. *Ecology* 85:2764–2771

Makino TT, Sakai S (2007) Experience changes pollinator responses to floral display size: from size-based to reward-based foraging. *Funct Ecol* 21:854–863

Thomson JD (1988) Effects of variation in inflorescence size and floral rewards on the visitation rates of traplining pollinators of *Aralia hispida*. *Evol Ecol* 2:65–76

400, Figure 5. Should these tests be adjusted for multiple comparisons?

434 and following. Again, I wonder if your dosing treatments are confounded with desiccation-related changes in pollen-transfer properties as the pollen is exposed to the elements. Treatment effects would still be important, but it might be misleading to attribute them to dosing *per se*.

442-444. I wrote a marginal note: “probably not a very meaningful variable.” Now I can't reconstruct my reasoning! Perhaps I thought that variances would be too high because complete removal would be hard?

452. Specify that you are considering narrow-sense DOL?

463. “Separate” might be too strong. Maybe talk of specialization toward or tendencies?

469-470. I could well believe that outer-anther pollen disappears because it has had more time to dry out, so it becomes more powdery/less sticky, and simply falls out of the anther by gravity (or is dislodged by jostling). I think this happens in penstemons.

488. Change “need” to “needs”

508-512. Add spaces to help the commas.

508. Might be worth noting that the strongest evidence for diminishing returns is related to grooming by corbiculate bees, where excess deposition triggers more grooming, and grooming removes grains from the active pool. Diminishing returns are probably general, but you shouldn't let readers get confused because your paper uses *Bombus* for experiments although the principal pollinator in nature is different.

514. Good point, which does recognize the particular nature of the pollinator.

656. What is “pavo 2”?

Appendix B

UNIVERSITY OF CALIFORNIA, SANTA CRUZ

BERKELEY • DAVIS • IRVINE • LOS ANGELES • RIVERSIDE • SAN DIEGO • SAN FRANCISCO

SANTA BARBARA • SANTA CRUZ

DEPARTMENT OF ECOLOGY & EVOLUTIONARY BIOLOGY
DIVISION OF PHYSICAL AND BIOLOGICAL SCIENCES
COASTAL BIOLOGY BUILDING
FAX 831-459-5353

SANTA CRUZ, CALIFORNIA 95060

Dear Editor,

Thank you for the detailed and thoughtful comments on our manuscript, “Darwin’s vexing contrivance: a new hypothesis for why some flowers have two kinds of anthers”. We are pleased to hear that you will consider a revised submission.

We have addressed your concerns and those of the three reviewers. The major changes to the manuscript are as follows:

- We more clearly present the suite of traits shown by plants that have heteranthery (as opposed to some other forms of anther dimorphism)
- We describe in more detail the division of labour hypothesis and its predictions, with additional citations
- We clarify the differences between the division of labour hypothesis and our pollen dosing hypothesis
- We provide more details about the analyses of correlated evolution of traits involved in heteranthery in *Clarkia*
- In order to accomplish the above, we edited the entire manuscript for conciseness and moved the colour reflectance figure to the supplement (formerly Fig. 5, now Fig. S4)

We strongly disagree that we are not able to falsify the division of labour hypothesis as it has historically been presented, and we disagree that division of labour and our pollen dosing hypothesis are not alternative explanations of heteranthery. I think the confusion stems from us not taking the space to thoroughly present the syndrome of heteranthery, not fully circumscribing the division of labour hypothesis and its predictions, and not clearly explaining the objective for each part of our study. Although heteranthery is often defined literally as “two kinds of anthers or stamens within the same flower”, historically it is only applied to flowers that show a certain suite of traits, with conspicuous anthers positioned centrally and inconspicuous anthers deflected peripherally. The division of labour hypothesis as presented in the literature refers to the two types of anthers specializing in pollinating versus feeding functions during the same bee visits, with bees preferentially harvesting pollen from conspicuous “feeding” anthers while inconspicuous “pollinating” anthers dust the bee with pollen in hard-to-groom places. While heterantherous flowers that conform to this division of labour often have pollen dosing due to gradual anther maturation or gradual pollen release (e.g., in buzz pollinated flowers), they are not using heteranthery *per se* for the pollen dosing. We are presenting a new hypothesis in which the heteranthery itself functions as a pollen dosing mechanism, and the anthers are not differentially specialized on pollinating versus feeding functions. Importantly, we believe this hypothesis can help explain why many heterantherous flowers do not show a clear divergence in anther function between pollinating and feeding. I hope that our revisions explain this more clearly.

We believe the manuscript is substantially improved, and that all matters raised during review have been carefully considered and addressed. We very much appreciate the detailed and constructive reviews of both the AE and three reviewers.

In the following sections, we respond (in black font) to Editor and Reviewer comments (in blue font). Line numbers refer to the version without tracked changes.

Sincerely,

Kathleen M. Kay, Associate Professor
Ecology & Evolutionary Biology

Associate Editor

Board Member: 1

Comments to Author:

I have received three reviews of your manuscript entitled “Darwin’s vexing contrivance: a new hypothesis for why some flowers have two kinds of anthers”. The recommendations made by the first two reviews were divergent and I requested a third review. Going out for a third review delayed the process, and I apologize for the extended wait time.

All referees found the topic to be interesting, and referee 1 was strongly positive about the manuscript. Referees 1 and 3 both had positive comments about the ambitious, multifaceted approach taken to understand the evolution and function of heteranthery in *Clarkia*.

However, the reports from referees 2 and 3 were quite critical and referee 2 in particular has flagged issues that raise questions about the overall objectives of the study. In particular, referee 2 points out that the division of labour and pollen dosing hypotheses are not mutually exclusive. This point is also made by referee 1. I agree with this comment. But, if we accept that pollen dosing and division of labour can operate simultaneously, there would seem to be a problem for the manuscript, which sets up the two hypotheses as “competing” with each other and that the series of analyses presented in the manuscript can distinguish the “alternative” hypotheses. This point was further driven home by referee 2, who points out that pollen dosing does not require heteranthery to operate – many non-heterantherous flowers engage in pollen dosing. Moreover, if we again accept that the two hypotheses are not mutually exclusive (and/or that the definition regarding the presentation of pollen under division of labour is too narrow – a point made by referee 1), at least one of the experiments does not seem to enable a test of the two alternative hypotheses. The pollen dosing experiment described on Pgs. 11 & 12 seems to be a test of the function of pollen dosing (gradual versus simultaneous pollen release) in promoting pollen export, not a test of pollen dosing versus division of labour.

Referee 2 has also flagged concerns with the analyses (comment #2) and referees 2 and 3 questioned different aspects of data interpretation (referee 2: comment #6, referee 3: final comment in the comments to authors section). I found these points to be persuasive, further weakening the conclusions of the study.

On balance, this manuscript has several strengths, the most notable of which is the multifaceted approach to the study of a topic of interest to a broad audience. However, it has several weaknesses that raise

questions about the general framework of the study and its conclusions. Addressing these weaknesses, particularly those flagged by referee 2, would appear to require a major reworking of the manuscript.

Reviewer(s)' Comments to Author:

Referee: 1

Comments to the Author(s)

I have attached a Word file. I see no point in anonymity.

This paper seeks to compare two alternative (and ostensibly competing, line 14) explanations for the evolution of heteranthy: division of labor (DOL) versus pollen dosing (PD). As it happens, I have a foot in each of these camps, in that I'm a minor coauthor on a paper that is thought to provide one of the more convincing arguments for DOL (Vallejo-Marín et al. 2009), but have also been involved in promoting PD as a widespread and potent selective force in floral evolution (Harder and Thomson 1989, etc.). Therefore, it seems simplest to waive anonymity.

This is an exemplary paper that offers a choice of explanations for an intriguing phenomenon, identifies an illustrative plant genus, and explores the evidence in a multi-pronged way that I would characterize as "using every blade on the Swiss Army knife." I find this an appealing and powerful approach. The writing is clear and efficient, and could be published virtually "as is" without embarrassment to the authors or the journal. Nevertheless, I do have some suggestions for some further development.

Admirably, the introduction gets right to the point, but I feel that the contrast between DOL and GRP needs more development. Specifically, the two hypotheses need to be stated in terms of the specific points of differentiation that will be used to distinguish them. At a couple of places later in the MS (e.g., lines 247-8) Kay et al. refer to "predictions of our pollen dosing hypothesis" in ways that left me a little surprised or unprepared. I did not think that those predictions—and their supporting rationale—had been adequately stated in the Intro. Therefore, I had to do some conceptual backtracking to satisfy myself about the reasoning behind the predictions. This is the wrong place in the paper for a reader to be working this out. I would find it more comfortable to have gotten a more explicit account, perhaps even a list of predictions, in the Introduction. The current Introduction seems to depend on the reader to infer, for example, the critical defining characteristics of DOL from little more than the name itself. I, for one, needed more hand-holding to fully understand the authors' arguments.

We have extensively revised the introduction to describe DOL in more detail. We have detailed the predictions of the two hypotheses and moved some of this language from the methods to the introduction.

The most important example of this shows up in line 66, where DOL between feeding and pollinating anthers is stipulated to occur "*during the same bee visit.*" If I infer correctly, that seemingly innocuous little phrase about timing is *essential* to their operational definition of DOL. I see it as a restrictive definition that gives rise to a "narrow-sense" view of DOL. One could imagine the words "division of labor" being interpreted more broadly. For example, if rewards gleaned from earlier-dehiscing anthers served to entrain return visits by bees that would move pollen from later-opening anthers, I could interpret that as a division of labor. More broadly, I think that I could view *any* differences in the timing of anther function as DOL in the sense of the English words, but Kay et al. would reject such interpretations. If these definitional matters can be attended to in the Introduction, the later portions of the paper will read more smoothly because readers won't be waylaid by niggles.

Division of labour as a hypothesis refers to the division of labor between pollinating and feeding functions and has always been described as happening during the same visit. We have more clearly explained this in the introduction with additional citations. I couldn't find evidence in the literature of the

phrase division of labour being applied to heteranthy in any other context. Regarding the possibility that early-dehiscing anthers might provide the pollen reward and promote return visits to later-dehiscing anthers that specialize in pollination, so that DOL happens over time – I suppose that is possible, but we don't see any evidence in *Clarkia* that later-dehiscing anthers are more specialized on pollination than early-dehiscing anthers. If early-dehiscing anthers increase return visits to later-dehiscing anthers without any specialization on feeding v. pollinating functions, that is consistent with our pollen dosing hypothesis. Unfortunately, we were not able to mark bees to test if this was happening.

Another biggish point is that I think that one could entertain a third hypothesis, that the two whorls of stamens might be optimized for working with different species of pollinators, i.e., heteranthy might be a way of producing a generalist floral morphology that can be well served by visitors of different sizes and proclivities. Of course, this too could be considered a variety of DOL, but I didn't think it's the kind of DOL that Kay et al. are stipulating.

That is an interesting idea, but we saw no evidence for this in the two species we worked with in the field. Moreover, although the anther whorls are positioned differently when the flower opens, the outer anthers move to the center of the flower upon dehiscence so that they are positioned similarly to the inner anthers (see Fig 5A, for example). Although we don't have a specific data set on the mechanics of anther contact with different types of bees, I am confident we would have noticed something like this (at least enough to investigate) after many flowering seasons in the field. Finally, if this was occurring, it would still not fit the DOL hypothesis as traditionally presented (pollinating v. feeding functions).

I'll try to amplify this point by going through the paper and flagging various items by line number (along with any suggestions for minor fixups).

14. Must these hypotheses be *competing*? What prevents both from operating simultaneously?

I hope this is clear with our more detailed presentation of the two hypotheses in the introduction.

43. The word "stumped" is evocative and effective, but might be too colloquial for non-native speakers?

Agreed. Substituted "perplexed"

65 and elsewhere. I infer that these anthers dehisce gradually along their length, thereby allowing gradual pollen presentation within an anther (hence measurements in mm at line 438). It would be good to clarify explain this more (unless I missed it).

Yes, they first open at the tip and then gradually dehisce along their length. I added this information to the study system description in the methods (line 119), and remind the reader when I explain how we measure the length of exposed pollen (line 312)

66. Here's the key phrase "during the same bee visit." Emphasize?

Yes, we more clearly explain this in the introduction.

68. Important but vague. Consider explaining "worse performance" in terms of what was actually measured.

We added the phrase "in terms of stigma penetration and pollen tube growth" (line 73-74)

88. *Must* DOL predict simultaneous pollen release? Why? References? (See general comment above.)

Might not DOL extend to the first whorl serving to attract and entrain site-faithful visitors, and the second whorl to placing pollen optimally on them when they come back?

Historically, DOL has referred to different anther functions during the same visit. We have added additional references for this. I suppose that DOL could function across time, but it would predict that purple (outer) pollen should be proportionally more involved in pollination. It is not.

93. Good: "both types exported." This is exactly the way to clarify the different propositions. Lines 92- 97 are excellent, clear statements. Still, they could be bolstered by explaining their rationales to pacify skeptics who aren't convinced by the unadorned assertions.

We left this statement here but expanded the introduction so that it doesn't seem to come out of the blue.

121. In what sense is “primarily” used and measured here? Purely by frequency of visits, or do other considerations factor in?

We are referring to frequency of visits. We don’t have quantitative data about the relative efficiency of the different pollinators at transferring pollen; however, we did see all putative pollinators actually touching the anthers and stigma.

125. Are “greenhouse and live” two categories of study, or a compound description of a single type? I think the latter, but the phrasing could be improved by rearrangement.

Replaced “greenhouse and live *Bombus*” with “greenhouse and flight cage” (line 132)

138. Better to change “while” to “although” here (and everywhere else where simultaneity is not the intended meaning).

Done

147. It’s a picky point, but I would also expect it with pollination by pollen-collecting masarid wasps...

Point taken, although as far as I know there are no pollen-collecting wasps visiting any *Clarkia*. Adding that detail here seemed distracting.

171. Define “actively dehiscing.” Gradual unzipping?

We added a definition “actively exposing pollen” (line 168). The epidermis of the pollen sacs doesn’t appear to be fused once the flower is open but rather is slightly overlapping like the front of a bathrobe. I don’t know if there is a term for that.

204. Did you make observations to ensure that drowning in bowls doesn’t disrupt pollen on bee bodies? And that it doesn’t affect pollen color? Based on my experience with a pollen dichromism in *Erythronium*, I also worry that color might be diluted or vanish entirely when grains germinate on stigmas. (Thomson, J. D. 1989. Germination schedules of pollen grains: implications for pollen selection. *Evolution* 43:220-223.)

We tried pollinations in the greenhouse with pure loads of white or purple pollen and they kept their color, even after being frozen and/or stored. Although the bees have scopae, the pollen seems to be almost as tightly packed as in corbiculae, such that there were solid clumps of pollen on the bees. If these clumps occasionally came off, I don’t think that would affect our overall proportions of pollen colors in the scopae, but could have led to some of the very low values for pollen loads. We also tried catching the bees live, but the pollen definitely came loose in the nets/aspirators/kill jars, so the bowls seem like the best option for catching the bees with limited disturbance.

220-221. Here’s one of the asserted predictions about the competing hypotheses that made me uneasy.

I hope this is better supported now with our revised introduction.

230 and surrounding lines. Obscure technical jargon, e.g., “boxcar width,” WTF? Consider relegating these details to an online supplement.

These are settings on the spectrometer that would be needed to replicate our results. I think it is important to add them to the main methods, but I tried to make this section clearer. If preferred, we can move them to the supplement.

238. Please explain the biological significance and behavioral implications of green contrast (here, not in a supplement).

We added a more detailed explanation here (lines 235-38). We found qualitatively similar results using total color contrast as calculated from Chittka’s hexagon distances, but for space decided to only present one metric.

246. Bees “only” notice the inner whorl?—this sounds too strong. Wouldn’t a partial preference suffice?

Agree. I changed “only” to “preferentially”

247. In my conception of pollen dosing, it would work perfectly well without any recognition by the bees—as it does in the many species that dose but are not heterantherous. I must be missing something.

That was awkward language. We had meant that bee visitation would respond to our treatments. We changed to “forage on”.

259. Capitalize Poisson, I think.

Done

279. “Free access” = what exactly? Ok, I see you treat this below. Presumably these were worker bees. (Actually, contrasting workers with males (non-pollen-seeking) would be an interesting idea.) Their motivation to collect pollen may depend pretty strongly on the number and hunger level of brood in the nest. If she is desperate for pollen, even a single worker could strip all of the available pollen from 15 flowers. Did you pay attention to the size of the bees? This can vary a lot in captive bees, depending on how they have been fed.

I clarified the language (lines 276-79), now specifying worker bees and that they had a sugar solution inside the hive. The worker sizes did vary, but we didn’t have a large enough sample size to examine how that might affect our experiments. The order of treatments was randomized, so I don’t think there was any systematic bias in bee size with treatment.

310-313. Again, the implications of “length of exposed pollen” would be easier to understand with more explanation earlier in the paper. This is a clever experiment, but I wonder if the contrast between just-starting-to dehisce anthers and fully-dehisced anthers is really adequate to isolate a pure dosing effect. I would worry about confounding with pollen age and desiccation, which is likely to affect stickiness, electrostatics, and transfer characteristics.

I have added details early on (line 119) about how the anthers dehisce lengthwise and reiterate that here (line 312). I don’t have data on how pollen characteristics change with age but have been working with these plants for several years, including doing many controlled crosses, and I haven’t noticed any obvious difference in pollen characteristics while the flowers are fresh (not wilted). If you let the flowers go unvisited for several days until they wilt, the pollen is still viable but has obviously lost its stickiness. We added the sentence, “Although flowers in the no dosing treatment were one day older (out of a 4-6-day male phase in the greenhouse), they were still fully turgid and the pollen appeared to have the same consistency.” (lines 309-311)

317. I am surprised that you found it feasible to weigh corbicular loads from such short foraging bouts. I would never have thought to try this. Even if you could get the pollen off as intact pellets, I would worry about variance induced by bee size, motivation level, etc. If you can give some extra detail about feasibility/repeatability, others might be encouraged to try this. Should you do more in the future, looking separately at both sides of the bee would give some insight into the repeatability question.

We have a nice scale, and it was possible to pull the entire pellet off with forceps under the dissecting scope. Since the pollen is purple, it was easy to see if we had removed the entire pellet. I think we still have the bees in the freezer, so I will see if I can compare both sides of the bee (but not for this paper!). I’m sure there was variation due to bees, but since we randomized the order of trials, I don’t think this would have biased our results.

343. In Figure 1: So, what genius taxonomist decided to name one of the no-heteranthery species *C. heterandra*? Might there be a story there?

John Torrey. As explained in the Fig 1 legend, *C. heterandra* has evolved to be autogamous and the inner anther whorl is sterile (perhaps a way to adjust the pollen:ovule ratio for selfing). However, there is no color difference between anther whorls, so we scored it as non-heterantherous.

352. Figure 2. Very nice. I see that at some point in the text that you introduce the term “color heteranthery.” It might be better to do that earlier, and it would be good to specify whether that’s what you mean by heteranthery in this table.

Now that we have more clearly presented the suite of traits involved in heteranthery, I took out the phrase “color heteranthery” because I thought it was distracting. However, we more clearly define how heteranthery was scored for the comparative analyses: “in this case strictly defined as a colour difference between the inner and outer anther whorls” (line 145-46)

360 onward. By contrasting the color ratios in the scopae to what’s on the stigmas, you seem to be implying that the scopal loads are entirely “feeding” pollen. This would be true for apid bees with corbiculae, but it’s not as clear-cut for bees with looser scopal loads. The grains in the scopae are probably still available for stigmatic deposit. This does not invalidate the importance of finding a difference

in the ratios of the two pools of pollen, but it might be worth pointing out, especially because the paper includes results from both corbiculate and non-corbiculate bees. There's some material on the differences here: Parker, A.J., J.L. Tran, J.L. Ison, J.D.K. Bai, A.E. Weis, and J.D. Thomson. 2015. Pollen packing affects the function of pollen on corbiculate bees but not non-corbiculate bees. *Arthropod- Plant Interactions* 9:197-203.

Thank you for the interesting reference. Although scopal pollen may still be available for pollinating, it should represent what the bees collected and groomed, and therefore what was intended for feeding. Importantly, there was no significant difference in the ratio of white to purple pollen ratios between the scopae and the stigmas. I hope this comes across more clearly now.

I also think the most common non-corbiculate pollinator in our study (*Hesperapis regularis*) may differ substantially from the Megachilid and Halictid bees studied in Parker et al. 2015 in how the pollen is packed. Whereas the bees studied in Parker et al. have loose, dry pollen loads, *H. regularis* glazes the scopal loads with nectar during foraging bouts, and the viscin threads of the *Clarkia* pollen cause it to densely clump together (see Portman, Zachary M., and Vincent J. Tepedino. 2017. "Convergent Evolution of Pollen Transport Mode in Two Distantly Related Bee Genera (Hymenoptera: Andrenidae and Melittidae)." *Apidologie* 48: 461–72. <https://doi.org/10.1007/s13592-016-0489-8>.) In practice, the clumps on the *Hesperapis* scopae did not look or behave much differently than the clumps on the *Bombus* corbiculae when we removed them for examination.

We added a sentence to the methods: "*H. regularis* glazes scopal loads with nectar so that they are densely clumped [31]" lines 209-210

369, Fig. 3. I like the paired panels in this figure.

Thank you!

370. In relation to the comment on line 360, perhaps the increase in purple pollen with total counts could be driven by visits that happened to result in particularly intimate contact with scopal pollen. It would be good to know a little more mechanical natural history about how bee size, bee behavior, and floral morphology interact to produce contacts and transfer.

The high purple proportions in high total counts is characteristic of both scopal loads and stigma loads, which shouldn't be the case if the purple anthers are not targeted for feeding. We are working on more detailed analyses of bee visitation and have a new high-speed video camera we have been using, but that is not ready for publication at this time.

379 and elsewhere. There's disagreement between Figure numbers in text and captions for 5 and 6. Looks like a former Fig. 4 was deleted?

I have carefully combed through the manuscript and fixed any discrepancies.

388 and following. Again, I am struck by the possibility that early rewards might influence later visitation rates (which might arguably be considered a kind of DOL). It would also be helpful to know more about how many flowers these plants produce, and over how many days? With bumble bees, at least, there is evidence that early-stage rewards by certain plants can confer higher visitation to later stages. I'm not up to date on the literature on such spatial-memory-holdover effects, but here are some older papers that I consider relevant. Bees that don't trapline might not show such effects.

Cartar RV by bumble bees: responses to in quality. –2771

TT, Sakai S (2007) to floral display size: from size-based to Funct Ecol 21:854–863

JD (1988) Effects of in inflorescence size and floral rewards on the visitation rates of of *Aralia* Evol Ecol 2:65–76

It is possible that this occurs. We were not successful in following individual *Hesperapis*, since they move erratically. The plants typically produce several flowers that open sequentially over weeks, from the base to the top of the indeterminate inflorescence. However, in that case, there would not be DOL as historically circumscribed because both anthers would be functioning as a bee reward, just spread out over time.

400, Figure 5. Should these tests be adjusted for multiple comparisons?

The significance levels come from a statistical model in which all treatments were compared to the control. Error bars represent 95% confidence intervals around treatment effects (indicated in the y-axis). 434 and following. Again, I wonder if your dosing treatments are confounded with desiccation-related changes in pollen-transfer properties as the pollen is exposed to the elements. Treatment effects would still be important, but it might be misleading to attribute them to dosing *per se*.

We added the following sentence to the methods (line 309-311): “Although flowers in the no dosing treatment were one day older (out of a 4-6-day floral lifespan in the greenhouse), they were still fully turgid and the pollen appeared to have the same consistency.”

442-444. I wrote a marginal note: “probably not a very meaningful variable.” Now I can’t reconstruct my reasoning! Perhaps I thought that variances would be too high because complete removal would be hard?

We thought that the amount of pollen the bees had collected was interesting because it indicates that for the no-dosing flowers, more pollen was lost from the system, as opposed to more pollen being groomed and collected by the bees. We did not see evidence that pollen availability (dosing v no dosing flowers) affected grooming behavior as in Harder 1990. The bees seemed to groom after almost every visit even if they didn’t pick up much pollen on their bodies.

452. Specify that you are considering narrow-sense DOL?

We now clarify how we consider DOL in the introduction. I don’t think it is fair to call it narrow-sense DOL because I couldn’t find any other references to DOL for anthers in the literature.

463. “Separate” might be too strong. Maybe talk of specialization toward or tendencies?

We replaced this word with “specialize on” (line 449)

469-470. I could well believe that outer-anther pollen disappears because it has had more time to dry out, so it becomes more powdery/less sticky, and simply falls out of the anther by gravity (or is dislodged by jostling). I think this happens in penstemons.

I agree this happens in *Clarkia*, but I have only seen desiccation in much older flowers. However, I think it does get dislodged by wind and jostling, and it is definitely eaten by beetles.

488. Change “need” to “needs”

Done

508-512. Add spaces to help the commas.

Oddly, this is the ProcB style.

508. Might be worth noting that the strongest evidence for diminishing returns is related to grooming by corbiculate bees, where excess deposition triggers more grooming, and grooming removes grains from the active pool. Diminishing returns are probably general, but you shouldn’t let readers get confused because your paper uses *Bombus* for experiments although the principal pollinator in nature is different.

We actually didn’t see a difference in *Bombus* grooming frequency with dosing v. no dosing flowers. These *Bombus* seemed to groom after almost every visit, even if the visit was to a female phase flower. I left this out of the manuscript because it seemed tangential to the goal of demonstrating that dosing by the outer anthers could increase pollen export over the flower lifespan. We are working on trying to replicate this experiment in the field with the native pollinators, and I think we will nail it during the next good flowering year. We are already over the space limit, and I would like to leave out a discussion of the differences between bee types, especially since I don’t really understand how grooming differs between them. I see *Hesperapis* groom, and it has densely clumped pollen loads on its scopae much like the corbicular loads. We can’t quantitatively compare its grooming frequency to *Bombus* because we haven’t similarly confined it to a flight cage.

514. Good point, which does recognize the particular nature of the pollinator.

656. What is “pavo 2”?

Pavo 2 is an R package for taking color reflectance data and modeling how it is perceived in different visual systems with known photoreceptor sensitivities.

Referee: 2

Comments to the Author(s)

In this study, Kay and colleagues investigate the evolution and function of differently coloured anthers in the *Clarkia*. The authors propose a hypothesis to explain the different colouration of anthers in these species, namely, selection for gradual pollen presentation. Below I provide some major and minor suggestions.

Major suggestions:

1) The definition of heteranthery (eg., abstract and line 37) used here needs to be refined (what qualifies as “types”? is a difference in colour sufficient?). Two morphologically different sets of stamens is not necessarily heteranthery, as commonly understood by plant evolutionary biologists. For example, stamens with different filament lengths in Brassicaceae or Scrophulariaceae are not considered heterantherous. This may seem like a small detail, but one could argue that it lays at the foundation of the thesis presented in this manuscript. If heteranthery is defined as any flower in which anthers are not morphologically the same, then it is hardly surprising that no single hypothesis can explain the evolution of what are certainly not evolutionary convergent structures.

Thank you for pointing out this lack of detail/context. I have revised the introduction to explain more clearly how heteranthery has traditionally been defined, including the suite of traits exhibited by heterantherous flowers (which differ from the different filament lengths in mustards and scrophs, as you point out). I also have more clearly explained the predictions of the division of labour hypothesis. I think this will help clarify why we really are able to set these up as competing hypotheses and reject division of labour for *Clarkia*.

2) The finding that heteranthery evolved once in *Clarkia* (line 329) raises some interesting questions about whether this is a good system for studying the evolution of heteranthery. With a single evolutionary transition it is probably not possible to meaningfully conduct statistical tests of character correlations between heteranthery and other features. Would it be possible to please estimate and provide in the text the number of evolutionary losses of heteranthery inferred in this study? From Figure 1 one can see a single gain and two losses of heteranthery. With no phylogenetic replication, it is possible to imagine that anther colour dimorphism in the flowers of some *Clarkia* do not necessarily serve the same function that caused the evolution of dimorphism in the first place.

Although heteranthery only evolved once, it has been lost twice in species that have evolved autogamy (selfing), so there are actually three transitions. I had reported the number of losses in the legend of Fig. 2, but now repeat that in the main text (line 327). Although there are only three transitions, the associations with bee pollination, anther movement, and gradual pollen release are consistent enough to be significant (or marginally significant, in the case of bee pollination) in the comparative tests we use (which are more powerful with increased number of transitions). Although this is not an ideal system for solely comparative work on heteranthery, we believe this aspect of the study helps support the more detailed mechanistic work on *C. cylindrica* and *C. unguiculata*. Moreover, the tight associations between color differences and both anther movement and staggered dehiscence directly support the pollen dosing hypothesis and contradict division of labor.

3) It is not clear how the new hypothesis proposed here (gradual pollen release) can explain the evolution of heteranthery. Gradual pollen release can be achieved through a variety of ways, and even heterantherous species show other strategies for gradual pollen release. Why has heteranthery evolved in species which already possess gradual pollen release? In which conditions would heteranthery evolve to dose pollen instead of other (simpler?) strategies such as gradual anther maturation, etc.

Gradual pollen release is known to be achieved through many routes (gradual anther maturation, gradual flower maturation within inflorescences, gradual inflorescence maturation) and is increasingly recognized

as an important route for increasing plant fitness. *Clarkia* does all of these things, too, but we are arguing that heteranthery makes the pollen dosing more effective and flexible. It is more effective because the specialized bee pollinators are very attuned to pollen rewards and they will go after any available pollen, even prying open the gradually-opening anthers further to dig the pollen out. The heteranthery adds an additional way to “hide” pollen from the bees with visual and positional crypsis and then gradually present it. It is more flexible than spreading the pollen across more flowers because the floral lifespan can speed up or slow down quickly in response to current weather conditions. So, the flower can lengthen the pollen presentation period when conditions are good, without having to make a separate flower, and shorten the period when conditions are too hot/dry.

Imagine a mutation arises that makes the outer anther whorl darkly colored with pigments similar to the petals/sepals. If this makes the gradual anther maturation more effective at dosing pollen, it would be favored and become common. Similarly, the style is deflected to the periphery when the flowers open, and then once the male phase is done, the style becomes erect and positioned at the center of the flowers. There could have been a genetic change that made the outer anther whorl behave similarly (start deflected and then become erect). These changes don't seem very complex to me because they are taking a trait that is already expressed in a different whorl of the flower and applying it to the outer anther whorl. Of course, we know nothing about the genetics of the heteranthery, so its simplicity is speculation.

4) In line 52-53 it is said that “it would be naïve to assume that bees, which have long coevolved to forage efficiently, will often overlook available pollen in inconspicuous anthers.” This may seem like an unfortunate wording as it implies that flowers cannot “fool” pollinators. It could be argued that there are plenty of examples of floral crypsis and deceptive pollination that show that pollinators indeed can be manipulated to visit flowers even when this is not necessarily the most efficient behaviour for the floral visitor.

I meant that the division of labour hypothesis may be naïve. I have removed this sentence. We are hypothesizing that the colour and positional crypsis work to protect the pollen in the outer anthers from early removal.

5) In line 101-104 the authors state that to test their new hypothesis they compare pollen export in plants with and without gradual dehiscence. However, one could argue that this experiment cannot possibly be used as a way to refute the division of labour hypothesis since gradual pollen presentation may not be a strategy that is mutually exclusive with the division of labour. Gradual pollen dispensing is likely favoured in most species of plants that receive a sufficient number of floral visits whether they are heterantherous or not.

You are correct that this experiment is not aimed at refuting division or labor but rather providing support for our hypothesis that the outer colored anther whorl is effectively dosing pollen in a way that would increase pollen export. We title this subsection “*Does gradual pollen presentation increase pollen export?*”) lines 299 & 419

6) A potential experimental problem is that pollen in the scopae was used to assess pollen export. Pollen in the scopae is removed from the pollination process and in many cases, particularly in some heterantherous species, the pollen transferred to stigmas is the one that is more likely to escape from bee packing in the scopae. The spatial arrangement of feeding and pollinating stamens might make pollinating anthers more likely to place pollen in parts of the bee's body which are harder to groom (safe sites) and thus the ration of pollen from the two anther types in the scopae is not a good reflection of pollen export. In fact, I would interpret the finding (lines 360-362) of less than expected purple, outer (“pollinating stamen”) pollen in the scopae as CONSISTENT with the division of labour hypothesis.

Pollen in the scopae was not used to assess pollen export, and I can't see where this comment comes from. Pollen on the stigmas was used to assess pollen export, and pollen in the scopae was used to assess bee collection/feeding. I think this is a misunderstanding. We have reworded the opening sentence of this part of the results to make it clearer (lines 356-57). Stigmas and scopae have statistically similar proportions of white and purple pollen, which shows that white and purple anthers are not differently specialized on pollinating versus feeding. The general underrepresentation of purple pollen in both sample

types, compared to relative pollen production by the anthers, shows that purple pollen gets lost disproportionately from the system. My guess is that it is eaten by florivores and sometimes knocked off with wind since it is exposed for so much longer than the white pollen.

Minor suggestions:

7) Line 25. Suggested change: “We find no support for division of labour in *Clarkia*, but multifarious...”
done

8) 157-158. Please clarify. Simultaneous presentation would refer to the two anther types offering some pollen at the same time. But this does not preclude gradual presentation of pollen (eg. gradual presentation of pollen as the flower ages)

I think this will make more sense with our revised introduction.

9) Line 166. Only one flower was studied per species? What is the variation among flowers of the same plant/species?

It was very difficult to get an uninterrupted series of time lapse photos of the same flower with consistent temperature, lighting, and humidity, since each flower lasts for several days. Thus, we chose to sample as many species as we could instead of multiple flowers per species. However, we grew more flowers of each species in the greenhouse, and our time lapse results for single plants seem consistent with what we saw those species generally doing in the greenhouse. Although the exact timing of each flower is sensitive to light/temp/humidity, heterantherous species always show staggered maturation of the anther whorls whereas non-heterantherous show no difference in timing (shown in Fig 3), so that variation would not qualitatively affect our results.

10) Line 177 and earlier. Please define colour heteranthery.

We have revised our initial description of heteranthery in the introduction, now more fully describing the suite of traits that have traditionally been called heteranthery, and have revised our language here (lines 145-46) to indicate that we are scoring species as having heteranthery when the outer whorl is differently colored than the inner whorl. Please excuse my American spelling of colour. I have fixed that throughout.

11) Lines 180-182. Could you provide more details of how the model was parametrised?

We have added more detail about the Pagel models in the supplement and made our language more precise in the main text (lines 175-185)

12) Line 183. Was this a different test (combination of traits) than above?

Yes, this was a test of colour polymorphism versus anther movement. We have now explained this more clearly. (Lines 183-85)

13) Line 184-186. Can you please provide more detail of how this model was constructed?

We have added more detail about the Pagel models in the supplement.

14) Does the pan trap with soapy water remove pollen from bees? This might be likely especially if bees stayed in the pans for the full collecting period (4 hours)

While it is possible that some pollen was removed from bees, most still had sizable clumps in their scopae and there were no visible clumps in the water. More importantly, even if some pollen was removed, it is very unlikely that the two colours of pollen were removed at different rates (since they were generally mixed together in clumps), so any pollen removal should have no effect on our analysis of colour proportions.

Referee: 3

Comments to the Author(s)

In this manuscript, Kay et al. studied *Clarkia* species to propose and validate a new hypothesis to investigate the function of heteranthery, an outcross pollination hypothesis, in contrast to the division of labour hypothesis. I am impressed with the amount of work conducted, and I enjoyed the multiple experiments and observations gathered to show that the two types of anthers seem to provide similar

functions (at least feeding bees). I am not an expert in heteranthy, but the division of labour hypothesis states that flowers with two types of anthers play a different role, one type for pollination, the other to feed bees, and pollinating anthers export more pollen than feeding anthers. In this study, the combination of observational/correlative data and experimental manipulations shows that heteranthy and associated traits evolved with bee pollination, and that the anthers classified as pollinating anthers are used by bees to collect pollen. They also show that anther movement and dehiscence functions to gradually deliver pollen, as opposed to full on presentation, in turn securing more visits and more pollen export in subsequent visits.

I enjoyed the question-approach of the authors to address specific functional hypotheses. However, the main flaw of the manuscript is the lack of clarity in the motivation of the study as exposed in the introduction, and the fact that the narrative used by the authors sounds “defensive” and focused on “our results don’t agree with the division of labour hypothesis but with the out-cross pollination hypothesis”. At present, it is necessary to read the introduction and the methods, and a lot of reading between the lines to gain a full picture of the two hypotheses and what the authors want to do: do you want to present a new way to understand how heteranthy evolved, or do you want to reject the division of labour hypothesis?. If the authors want to follow the narrative of conducting experiments and presenting data to see if it validates one hypothesis or the other, I suggest to present the two hypotheses at the introduction, and move the bits in the M&M that sound like “oh and by the way, under this hypothesis the expectation is...”. I have added my comments in the text where this happened. So it is important to present the division of labour and outcross-pollination hypotheses in the introduction, and hence create specific aims (already there in the different questions presented in the M&M) in relation to specific components of the hypotheses or tests to validate one or the other. What I gathered is that the authors want to demonstrate that heteranthy in *Clarkia* evolved in the context of the promotion of cross-pollination, as opposed to division of labour. Focus your narrative on that, and then you will be able to create a stronger argument for an alternative hypothesis to the division of labour.

Thank you for this feedback. We have substantially revised the introduction to better define heteranthy and the predictions of the division of labour hypothesis. We left the paragraph in the introduction that details the differing predictions and the reasoning behind each piece of work largely intact. We didn’t mean to sound defensive in the methods and results, we were just trying to remind the reader about different predictions at the beginning of each section, since this is such a multifaceted study. I would like to leave most of these phrases in. Before we added them, every friendly reviewer got lost in the methods/results.

After reading the manuscript, I was left with mixed feelings about one hypothesis or the other. What is unclear to me is why two types of anthers evolved. Division of labour is set under the context of male to male competition and intrafloral male competition. Intrafloral male to male competition doesn’t make much sense to me because it doesn’t benefit the total male fitness that a single flower can achieve.

I wouldn’t agree that Division of Labor posits intrafloral male to male competition. Rather it is a way for anthers to specialize on different functions and reduce overall pollen loss to bees. Feeding anthers provide a reward to bees while pollinating anthers specialize in exporting pollen. I think this was a misunderstanding. Pollen presentation schedules evolve under indirect male-male competition for access to mates, but this competition occurs among plants not within a flower. I tried to simplify the language in this paragraph to avoid this misunderstanding (paragraph starting at line 486).

Instead, a strategy that increases pollen export and siring success would be selected (so that male gametes of a flower don’t compete, or compete less). This is where the advertisement role of inner anthers is so interesting. The anther removal experiment is really inspiring to learn how much bees rely on those anthers to perceive and visit flowers, and whether they function as advertisement so that bees learn where

to go later in the afternoon, and repeatedly as anthers open and deliver pollen. Note that plants where the division of labour has been tested also release pollen gradually. Having a set of anthers that increases visitation later in the day is a good way to increase pollen delivery and individual (flower level) male fitness, instead of intrafloral male to male competition. It would be interesting to know if the fertility of pollen from both anther types is the same, or whether inner anthers are less fertile. Then you would have a new twist in the division of labour hypothesis to promote cross-pollination.

It is possible that inner anthers allow bees to learn where the flowers are so that pollinator visitation is increased later when the outer anthers are actively dehiscing. We were not able to mark individual bees in the field to track learning, but in Figure 5, you can see that removing inner anthers when flowers opened slightly reduced visitation in the afternoon. This could be because bees learned that those specific flowers were not rewarding, or it could be because there was a lack of residual inner anther pollen, or it could be because the flowers were less visually attractive with those with anthers missing (even after the pollen has been removed, the shriveled anthers provide some visual contrast in the center of the flowers). Importantly, though, if the inner anthers are increasing later visitation to the outer anthers, this is much more consistent with heteranthery as a gradual pollen dosing strategy than division of labour, in which the reward is primarily provided by the central feeding anthers while pollination is achieved by the deflected and visually cryptic anther. I hope the distinction between our pollen dosing hypothesis and division of labour is more clear now that we have revised the introduction. Finally, as we stated in the introduction (lines 71-74) and discussion (lines 454), a recent study has found that the inner anther pollen is slightly more fertile than the outer anther pollen, which also contradicts them being feeding anthers.

In appendix S1, the AIC values of the models (heteranthery depends on pollination -I would say correlated with, or evolved after-) is similar (ca. 2 units larger) that a model with characters are independent and pollination depends on heteranthery...so even the first model shows the lowest AIC, it is still close to the AICs displayed by other models. I am not fully convinced that correlated evolution tests are helping here to demonstrate what are the most likely evolutionary events. It is clearer in the tests of correlated evolution between anther movement and colour of anthers. Also, use a narrative that infers correlation and not causation.

We used the wording “depends on” in a statistical sense, since the models use different dependent and explanatory variables. We have described these models and the general Pagel method more thoroughly in the supplement, made our language more precise in the main text, and removed the “depends on” wording. With only three transitions in color heteranthery, we thought this marginally significant result of the correlation with pollination state was worth reporting. However, we have tempered our language in the results (lines 331-32). More importantly, we have striking statistical relationships between heteranthery and anther movement and between heteranthery and staggered dehiscence across anther whorls, which suggest that heteranthery is commonly a mechanism of pollen dosing across this clade of *Clarkia*. There are no heterantherous species in our study without anther movement or staggered dehiscence between anther whorls. This suggests that heteranthery did not first evolve as a division of labour strategy and then change to a way of dosing pollen.

I have added specific comments in the text.

Thank you. We have incorporated all of these comments except for combining the first two paragraphs. With our edits to the introduction, we felt these were better as two separate paragraphs.

Appendix C

UNIVERSITY OF CALIFORNIA, SANTA CRUZ

BERKELEY • DAVIS • IRVINE • LOS ANGELES • RIVERSIDE • SAN DIEGO • SAN FRANCISCO

SANTA BARBARA • SANTA CRUZ

DEPARTMENT OF ECOLOGY & EVOLUTIONARY BIOLOGY
DIVISION OF PHYSICAL AND BIOLOGICAL SCIENCES
COASTAL BIOLOGY BUILDING
FAX 831-459-5353

SANTA CRUZ, CALIFORNIA 95060

Dear Editor,

We are delighted about the acceptance of our manuscript, “Darwin’s vexing contrivance: a new hypothesis for why some flowers have two kinds of anthers”.

We have addressed your and Reviewer 3 comments in detail below, with comments in blue and our responses in black. Line numbers refer to the version without tracked changes.

We hope you will find this revision satisfactory, and we thank you again for a productive review process.

Sincerely,

Kathleen M. Kay, Associate Professor
Ecology & Evolutionary Biology

AE Comment

L219 - For the study described as “Are outer anthers cryptic, and do they become conspicuous to bees as they release pollen?” indicate the number of flowers sampled per species, and the stage at which flowers were sampled (before, or during pollen dehiscence of the inner and outer whorls of anthers).

We added this information on Line 222.

Reviewer 3 Comments

A key aspect to test the hypothesis of division of labour vs. pollen dosing is to provide good estimates of male fitness. But the male fitness proxies are inferred from bee behaviour and pollen collection, and through pollen deposition on the stigmas. With regards the later, this is true since what limits male fitness is access to partners, and in this manuscript is assessed by counting pollen load on the stigmas. But the authors showed that the two types of anthers don’t seem to differ (too strongly) on the amount of pollen deposited on stigmas (at least in one of the experiments). Access to partners can also be quantified by the % flowers with pollen from one or the other anther type. And indeed that could be (potentially) a better proxy of male fitness. The amount of pollen on the stigmatic surface depends on the total stigmatic area, which could (also potentially) express variation among flowers (this is in relation to line 856 and line 1002)

Male fitness is notoriously difficult to quantify, often requiring laborious genotyping of progeny. One of the really nice aspects of the *Clarkia cylindrica* system is that the pollen from the two anther whorls is differently coloured, such that we can quantify the fate of the pollen without genotyping. Under the division of labour hypothesis, we expect the outer anther (purple) pollen to be disproportionately represented on stigmas if those are the “pollinating” anthers (compared to the white pollen, which should be consumed by bees). Instead, we find white pollen is the most common pollen type on stigmas. This is about as direct a measure of the relative male fitness of the inner v outer anthers as one could hope for.

As for the % of flowers with pollen from one or the other anther type being a better response variable, I strongly disagree. Nearly all stigmas had a mix of pollen types, as one can see in Fig 4. All stigmas had at least some white inner anther pollen, and all but one stigma had at least some purple outer anther pollen. If we used the % of flowers as a response variable, we would be comparing 100% to nearly 100% (and even by that measure, the purple pollen is not more successfully exported). Since we used the *proportion* of purple pollen as our response variable, our results should not be sensitive to variation in stigma size among flowers. However, because we plotted individual data points on Figure 4, this alternative response (% of flowers with a certain type of pollen) can be reconstructed from the graph if any readers have a similar curiosity. There is simply no evidence that outer anthers are specialized on pollination, compared to inner anthers.

We also compare a proxy for male fitness between dosing and no dosing anthers of *C. unguiculata* in our captive bee trials. In this case, we are not working with a colour polymorphism because we are only working with the outer anthers. However, we only had one male flower type per array (either all dosing or all no dosing). Thus we could compare the amount of pollen deposited on the stigmas of the female flowers as a way to compare male fitness. Here (Fig 6B) we did not find a difference. However, we infer that the dosing flowers would have higher male fitness because they had pollen remaining for future bee visits, whereas the no dosing flowers were stripped bare.

I don't understand the reference to line 856 and line 1002 in this comment, but I hope I have clarified this general issue.

The experimental manipulation with *Bombus* is extremely smart and clearly laborious. But I wonder to what degree the results obtained with *Bombus* are comparable to those retrieved using native bees. I assume that *Bombus* and *Hesperapis regularis* are different in size, behaviour etc so that I am not entirely sure if the magnitude of the results obtained in the experiments involving two different bee species are comparable. Probably a qualitative comparison is possible, but it would be, at least good to acknowledge this. After all, heteranthery has evolved (and functions) in the context of local pollinators.

This is an understandable concern and is why we used wild plants and pollinators in this project whenever possible. Most bee-pollinated *Clarkia* receive at least some visits from bumblebees, including *C. unguiculata*, which we used in the trials with captive bumblebees. Of course, we did not use a local species of bumblebee, but rather used one that is commercially available. Over the past several years I have spent many hours observing the native pollinators and the captive *Bombus*, and I am convinced that there are no major qualitative differences in how they interact with the heteranthery. The native pollinators range in size, but that range overlaps with *Bombus impatiens*. They show similar frequencies of pollen collecting v. nectaring behaviour. They land on the center of the flower in the same way. They ignore undehisced outer anthers in the same way. I think the main difference is that *Bombus* is more methodical about moving up an inflorescence than *Hesperapis*. However, the native *Bombus* are similar to the captive *Bombus* in this regard. We also know that bee visual systems do not vary extensively, so I think the colour analyses are valid. We already acknowledge this issue in the text (lines 274-75). Lastly, we see a similar heteranthery phenotype, with staggered dehiscence and anther movement, across the

entire clade of heterantherous *Clarkia*, even though these are collectively pollinated by a variety of bees. I believe this indicates that the function of heteranthy is similar with different types of bees.

Line 756: I wonder if some of the patterns identified can be explained because native bees learn to search based on their experience. So for example, export of pollen from outer anthers could increase if bees had a positive experience in the same flower with inner anthers, and they could simply learn “where the good flowers for pollen are”. Is there a correlation between pollen production between inner and outer anthers? Although this calls for a quantitative (not qualitative) comparison.

I think a more parsimonious explanation for the increase in proportion of purple pollen (in scopae and exported to stigmas, Fig 4) with the total pollen load is that outer anthers produce more pollen but are more variable in retaining that pollen over the floral lifespan. So, some visits to late-stage flowers result in large pollen loads with relatively higher frequencies of purple pollen (although still less than expected based on pollen production). Either way, this doesn't directly bear on the main result here: the inner v. outer anthers are not specialized on feeding v. pollinating functions, and I don't think we have the data to speculate about bees remembering good flowers.

Line 813-819: This is precisely my point mentioned above. You might detect a signal if you estimate the number of flowers with pollen from each anther type, instead of quantifying pollen on the stigmas. The former is also a way to quantify access to partners (access to partners in terms of different individual flowers as well as ovules in one flower).

I don't understand the line number references. Perhaps the line numbering got reset somehow? Regardless, I believe I answered this criticism above. Nearly all flowers have a mix of pollen types, so quantitative measures of pollen deposition are more informative than the % of flowers with a certain type.